
# Long-term multi-source data analysis about the characteristics of aerosol optical properties and types over Australia

Xingchuan Yang[1], Chuanfeng Zhao[1]*, Yikun Yang[1]

1. State Key Laboratory of Earth Surface Processes and Resource Ecology, and College of Global Change and Earth

System Science, Beijing Normal University, Beijing, China

*Correspondence to*: Chuanfeng Zhao (czhao@bnu.edu.cn)

**Abstract.** The spatiotemporal distributions of aerosol optical properties and major aerosol types, along with the vertical

distribution of major aerosol types over Australia, are investigated based on multi-year AERONET observations at nine

sites, the Moderate Resolution Imaging Spectroradiometer (MODIS), Modern-Era Retrospective analysis for Research

and Applications, Version 2 (MERRA-2), Cloud-Aerosol Lidar with Orthogonal Polarization (CALIOP), and back-

trajectory analysis from the Hybrid Single Particle Lagrangian Integrated Trajectory (HYSPLIT). The annual aerosol

optical depth (AOD) at most sites showed increasing trends (0.002-0.028 yr$^{-1}$) except for that at three sites of Canberra,

Jabiru, and Lake Argyle, which showed decreasing trends (-0.004 - -0.002 yr$^{-1}$). In contrast, the annual Ångström exponent

(AE) showed decreasing tendencies at most sites (-0.044 - -0.005 yr$^{-1}$). The results showed strong seasonal variations in

AOD with high values in the austral spring and summer and relatively low values in the austral fall and winter, and weak

seasonal variations in AE with the highest mean values in the austral spring at most sites. Spatially, the MODIS AOD

showed obvious spatial heterogeneity with higher values appeared over the Australian tropical savanna regions, Lake

Eyre Basin, and southeastern regions of Australia, while low values appeared over the arid regions in western Australia.

Monthly averaged AOD increases from August to next austral spring peak (typically December-January), and decreases

during the March-July. Classification of Australian aerosols revealed that the mixed type of aerosols (biomass burning

and dust aerosol) are dominated in all seasons at nine sites, followed by biomass burning aerosol and dust aerosol. The

MERRA-2 showed that carbonaceous over northern Australia, dust over central Australia, sulfate over densely populated

northwestern and southeastern Australia, and sea salt over Australian coastal regions are the major types of atmospheric

aerosols over Australia. The CALIPSO showed that polluted dust is the dominant aerosol type detected at heights 0.5 - 5

km during all seasons. Australian aerosol has similar source characteristics due to intercontinental transport of aerosols

over Australia, especially for biomass burning and dust aerosols. However, the dust-prone characteristic of aerosol is

more prominent over the central Australia, while the biomass burning-prone characteristic of aerosol is more prominent

in northern Australia.



# 1 Introduction

Aerosols play a crucial role in the Earth's radiation budget and climate change both through direct interaction with the solar radiation, and through indirectly modifying the optical properties and lifespan of clouds (Albrecht, 1989; Charlson et al., 1992; Garrett and Zhao, 2006; Ramanathan et al., 2001; Twomey, 1977; Zhao and Garrett, 2015; Zhao et al., 2018). In addition, elevated aerosols degrade the air quality and visibility, causing adverse impact on human health and the environment (Luhar et al., 2008; Pope et al., 2009; Zhao et al., 2019; Yang et al., 2018). Therefore, the influential effects of aerosols at both regional and global scales cannot be ignored.

Australia is situated between the Pacific Ocean and Indian Ocean in the Southern Hemisphere with 70% of its land surface occupied by arid and semi-arid regions. The northern Australia is tropical savanna ecosystems with dense grasslands and scattered shrubs or trees. Frequent fires occurred during the dry season (April-November) each year in this region have released large quantities of biomass burning aerosols into the atmosphere (Radhi et al., 2012; Paton-Walsh et al., 2004). Schultz et al. (2008) reported that Australian fires contribute almost 8.25 % of global carbon emissions. Mitchell et al. (2013) demonstrated that Australia is a globally significant source of biomass burning aerosol from savanna burning. To the south of the savanna are huge desert regions (including Great Victoria, Simpson, Gibson, and Sturt deserts) and agricultural lands that have been under the control of prolonged drought, which makes Australia a dominant dust source in the Southern Hemisphere (Mitchell et al., 2010). Tanaka and Chiba (2006) found that the Australian annual dust emission is about 106 Tg in contrast to 1087 Tg from North Africa and 575 Tg from Asia. Luhar et al. (2008) demonstrated that the mean top of atmosphere (TOA) direct radiative forcing significantly increased during the 2004 burning season. Small et al. (2011) found that cloud fraction initially increased by 25% with increasing aerosol optical depth, followed by a slow systematic decrease (~18%) with higher aerosol optical depth. Hence, Australian aerosol constitutes a significant component of the global aerosol budget, with great impacts on regional and global climate and radiation budget.

There have been numerous studies that have focused on aerosol optical properties in Australia. The northern


Australia is highly influenced by biomass burning aerosol from wild fires during the dry season, while the central Australia

is mostly affected by dust aerosol (Yoon et al., 2016; Qin and Mitchell, 2009). Australian aerosol optical depth (AOD)

generally increases from the austral winter to spring, then decreases from summer to fall (Mitchell et al., 2017). Moreover,

Mitchell et al. (2017) reported a long-range coherence in the aerosol cycle over Australian continent due to the similar

source characteristics and intercontinental transport. Many studies found significant decreasing trends in AOD by using

ground-based or satellite remote sensing data (e.g. Cloud-Aerosol Lidar and Infrared Pathfinder Satellite Observations

(CALIPSO), MODIS, Multiangle Imaging Spectro Radiometer (MISR)) over Australia (Mehta et al., 2018; Mehta et al.,

2016; Yoon et al., 2016). There are also a lot of studies focusing on short-term aerosol properties during environmental

events, such as dust storms (Shao et al., 2007) or wildfires (Radhi et al., 2012; Mitchell et al., 2006), and most of these

studies are based on a specific site/region. In addition, aerosol properties in Australia have also been studied in a number

of field campaigns such as the Mildura Aerosol Tropospheric Experiment (MATE 98) (Rosen et al., 2000), the Biomass

Burning and Lightning Experiment Phase B (BIBLE-B) campaign (Takegawa et al., 2003), the Savannah Fires in the

Early Dry Season (SAFIRED) campaign (Mallet et al., 2017), and the Measurements of Urban, Marine and Biogenic Air

(MUMBA) campaign (Paton-Walsh et al., 2017). In previous studies, fire season has often been the focus study period

due to the significant impacts of biomass burning aerosol. Meanwhile, most of previous studies have focused on aerosol

properties at a specific site/region or short-term variations of aerosols due to the difficulty of obtaining ground-based

aerosol data (Mitchell et al., 2010; Bouya and Box, 2011). However, it is essential to characterize the national-scale and

long-range aerosol optical properties, which could provide a better understanding of the aerosol characteristics and their

impacts on climate over Australia.

Ground-based observations can provide more accurate data to estimate aerosol properties, while remote sensing

technique provides a better understanding of aerosol properties at large scale. In this study, multi-year aerosol optical

properties obtained from nine ground-based sun photometers, along with the MODIS AOD product over Australia are

analyzed to investigate the long-term spatiotemporal variations of aerosol optical properties over Australian continent. In addition, MERRA-2 AOD and CALIPSO aerosol profile data are used to characterize the spatiotemporal and vertical variations of aerosol properties in the atmosphere over Australia, respectively.

The paper is organized as follows. Section 2 describes the AERONET sites, data and methods. Section 3 illustrates

the spatiotemporal variations in aerosol properties, the classification of aerosol types and their seasonal distributions, and discusses the seasonal variations of aerosol vertical distributions in Australia. Section 4 summarizes the findings of the study.

## 2 Sites, data and methodology

### 2.1 Sites

Australia is located in Oceania between the Indian Ocean and South Pacific. Australia's topography is not too varied, most of which is desert plateau (Fig. 1a). Western Australia is covered by great deserts and large plains with scattered shrubs and grasses. The central Australia is the Central Lowlands, including the Great Artesian Basin (GAB). Eastern Australia is mainly composed of the fertile plains along the eastern coast and the eastern highlands with dense trees and grasses (Fig. 1b). Northern Australia is covered by tropical savanna with dense grasses, scattered trees and grassy

woodlands. In this study, nine AERONET sites in Australia were used for analysis. Table 1 shows the site locations and the available observation data period at each site. The Learmonth site is located in the west coast of Australia and it is situated on the North-Western dust pathway from the Australian interior. Lake Lefory is located in the southwest of Australia and next to the Great Victoria Desert. Lake Argyle and Jabiru are located in northern Australia, where biomass burning is a significant source of aerosols during the dry season. Birdsville and Fowlers Gap are located in central

Australia, where are exactly the dust transport path. Lucinda and Adelaide Site 7 are located in the eastern and southern coast of Australia, respectively. The Canberra site is located at the top of a hill in southeastern Australia. Note that the data available time periods are different among these sites.



### 2.2 Data

#### 2.2.1 AERONET data

The AERONET (AErosol RObotic NETwork) is a ground-based remote sensing aerosol network which measures direct solar and diffuse sky radiance in the spectral range of 0.34-1.02 mm (Holben et al., 1998). AERONET provides high quality measurements of aerosol optical, microphysical and radiative properties every 15 min. The uncertainty of the AOD is approximately 0.01 to 0.02 (Eck et al., 1999). In this study, level 2.0 quality controlled and cloud screened data of AOD, Ångström exponent (AE), fine mode fraction (FMF), single scattering albedo (SSA) are adopted. Because

the AERONET quality assurance level of 2.0 is not available during 2018-2020 at a few sites, level 1.5 data are used from 2018 to 2020 for those sites instead.

#### 2.2.2 MODIS data

    The Moderate Resolution Imaging Spectroradiometer (MODIS) instrument is a multispectral sensor with 36 bands between 400 and 1440 nm onboard the Aqua and Terra platforms (Remer et al., 2005). MODIS provides two kinds of

long-term aerosol products, the Level-2 daily products at 10 km and 3 km resolutions, and Level-3 daily, eight-day, and monthly products with a $1° × 1°$ horizontal resolution. Levy et al. (2013) showed that the expected errors of the L2 MODIS AOD product are about $±(0.05+15\%)$ over the land. In this study, the data record named "Deep_Blue_Aerosol_Optical_Depth_550_Land_Best_Estimate" from Aqua MODIS Collection 6 daily AOD product (MYD04_L2) with 10 km spatial resolution from July 2002 to May 2020 is used to examine the spatio-temporal variations

of aerosol optical properties in Australia.

#### 2.2.3 MERRA-2 data

    MERRA-2 is the NASA's newest global atmospheric reanalysis product. It is produced by the Global Modeling and Assimilation Office (GMAO), using the Goddard Earth Observing System (GEOS-5) atmospheric data assimilation system. The global natural and anthropogenic aerosols are simulated in MERRA-2 with the Goddard chemistry, aerosol,



radiation and transport model (Randles et al., 2017). MERRA-2 assimilates aerosol optical depth (AOD) from various

ground and remote-sensed platforms, which includes direct AOD measurements from AERONET, bias-corrected AOD

retrievals from the Advanced Very High Resolution Radiometer (AVHRR) instruments, and AOD retrievals from the

MISR and the MODIS over bright surfaces (Buchard et al., 2017). Buchard et al. (2017) demonstrated that MERRA-2

has high reliability in simulating aerosol properties. In this study, monthly AOD at 550 nm from MERRA-2

(MERRA2_400.tavgM_2d_aer_Nx) for total aerosols and different aerosol species during the period from July 2002 to

May 2020 are used to analyze their spatio-temporal variations.

**2.2.4 CALIPSO data**

The CALIPSO satellite was launched on April 28, 2006. It is a dual-wavelength (532 and 1064 nm) polarization

lidar designed to measure the global vertical profile of aerosols and clouds (Winker et al., 2003). CALIPSO provides

vertical profiles data of the attenuated backscatter at 532 and 1064 nm and the perpendicular polarization component at

532 nm during both day and night. In this study, globally gridded monthly product derived from CALIPSO Version 4.2

Level 3 aerosol profile product (CAL_LID_L3_Tropospheric_APro_CloudFree-Standard-V4-20) is used to investigate

the vertical distributions of different aerosol species in Australia. The data has 208 layers with 60 m vertical resolution

for heights up to 12.1 km above mean sea level.

**2.2.5 ERA-5 data**

ERA-5 is the latest atmospheric reanalysis data developed by the European Centre for Medium Weather Forecasts

(ECWMF). ERA-5 has a great improvement in horizontal and vertical resolutions in comparison to ERA-Interim and

covers the period from 1979 onward (Albergel et al., 2018). In this study, monthly ERA-5 data, including the U wind, V

wind, and precipitation were obtained from the Copernicus Climate Change Service (C3S).



### 2.3 Methodology


The AERONET observations, including the total, fine, and coarse-mode AOD at 500 nm, AE at 500 nm, FMF at 500 nm, and SSA at 440 mm, and the MODIS AOD products are utilized to examine the spatio-temporal variations of aerosol properties over Australia. MERRA-2 and CALIPSO AOD products are analyzed to present the spatio-temporal and vertical variations of different aerosol species over Australia. Different aerosol types have diverse optical, physical,

chemical properties with different impacts on the atmosphere (Charlson et al., 1992). Furthermore, the classification of aerosol can provide a key clue to the leading source of regional aerosols. Thus, two methods from Kaskaoutis et al. (2007) and Giles et al. (2012) are adopted to distinguish aerosol types by using the AOD at 500 nm, AE at 440-870 nm, and SSA at 440 nm, as illustrated in Fig. 2. As a result, five prominent types of aerosols such as biomass burning, clean marine, urban/industrial, dust, and mixed type aerosol can be obtained. Back trajectory analysis is conducted to identify the main

transport pathways of air masses. The 72-h back trajectories at an altitude of 500 m above ground level (a.g.l.) are simulated by using the Hybrid Single Particle Lagrangian Integrated Trajectory (HYSPLIT) model (Draxler and Hess, 1998). Moreover, the back trajectories are calculated every 6 hours a day at nine AERONET sites during the period 2005-2020.

### 3. Results and discussion

### 3.1 Spatio-temporal variation of aerosol optical properties

### 3.1.1 Long-term variations and trends of aerosol optical properties

The annual variations and trends of AOD, AE at the nine sites over Australia are shown in Fig. 3. The annual means of AOD were 0.06-0.15 over Australia, which indicates the general excellent air quality over Australia. It showed an

increasing tendency in the annual mean AOD at most sites in Australia during the observation period except for the sites of Canberra, Jabiru, and Lake Argyle, at which the annual mean AOD showed a decreasing trend of -0.004±0.033 yr$^{-1}$, -



0.002±0.035 yr$^{-1}$, -0.004±0.057 yr$^{-1}$, respectively. In particular, the highest AOD (0.20) at Canberra was observed in 2003

followed by a decrease to 0.06 in 2004. This is mostly related to the wildfires of southeastern Australia in January 2003,

which generated large amounts of smoke aerosols, leading to a maximum in AOD during that observation period (Mitchell

et al., 2006). Among the six sites with the increasing trend of AOD, Birdsville is with a more evident AOD increasing

trend of 0.028±0.065 yr$^{-1}$ during the period 2013-2020. It is worth mentioning that significant increasing trends of AOD

are observed during the period 2019-2020 at most sites, such as Adelaide Site 7, Birdsville, Fowlers Gap, and Lucinda.

The increasing trend in AOD is related to the frequent fire activities in Australia from September 2019 to January 2020.

In addition, optical and physical properties of aerosols during the Australia wildfires in 2019 will be discussed in detail

in our future study. The annual means of AE were 0.74-1.31 over Australia. The annual AE showed decreasing trends at

most sites during the observation period except for Lake Lefory, at which it showed an increasing trend of 0.013±0.216

yr$^{-1}$. This result indicates that the size of aerosol increased at Lake Lefory, while the size of the aerosol decreased at other

sites. The annual AOD and AE at Jabiru (0.15; 1.20) and Lake Argyle (0.14; 1.27) were higher than those at other sites

(0.06-0.10; 0.74-1.13) in Australia, which can be explained by the extensive and frequent wildfires over the tropic North

of Australia. Moreover, the annual AOD and AE at Jabiru and Lake Argyle presented significant interannual variations

in amplitude. Similar results have also been reported by Radhi et al. (2012) and Mitchell et al. (2013). This is likely a

consequence of the relatively high rainfall during the wet season and large smoke emissions during the subsequent burning

season. The rainfall likely suppressed the smoke emissions due to the high level of moisture in the air, but it also promoted

the growth of vegetation, leading to an increase in smoke emissions during the subsequent dry seasons (Mitchell et al.,

180    2013).

Spatial distribution of annual averaged (Fig. 4 (a-s)) and multi-year averaged (Fig. 4 (t)) MODIS AODs for the years

of 2002-2020 indicates that, overall, the patterns of AOD spatial distributions were similar from 2002 to 2020. However,

the magnitude of the spatial AOD distributions varied to some extent. It is worth mentioning that the years of 2002 and





2020 only covered July to December and January to May due to observation data availability, respectively. Figure 4

shows that the average annual AOD values in central and eastern Australia were higher than that in the western Australia.

This similar spatial variability of aerosols was also captured in previous studies (Mehta et al., 2016, 2018). AODs with

values smaller than 0.04 were found in the arid zone of western Australia, which was the sparsely populated area with

low aerosol emissions. The highest AOD values were found in tropical and subtropical zone with dense grasses or forests,

where AOD average value was greater than 0.1. Furthermore, the AOD was high over the central plains with 19-year

averaged AOD ranging from ~0.02 to ~0.14, which was mostly affected by the dust storms. It is clear that the temporal

trends in AOD estimated by AERONET and MODIS were generally consistent. Furthermore, the time series of annually

averaged MODIS AOD over Australian continent and eight main administrative regions during the period 2002-2020 are

shown in Fig. 5. The time series of the yearly averaged AOD over Australia showed a quite small decreasing trend (-

0.0003) during the period 2002-2020, but there were still short-term annual variations. Mehta et al. (2016) also reported

a decreasing trend in AOD over Australia from 2006 to 2016 by using CALIPSO data. Although the data for year 2002

and 2020 were incomplete and cannot represent the statistical characteristics of AOD throughout the year, the significant

impact of environmental events (e.g. wildfire, dust storm) on Australian aerosol can be found from 2002 to 2020. In

southeastern Australia, the Australian Capital Territory and Victoria had similar annual variation characteristics in AOD,

while the New South Wales, Queensland and South Australia had their similar annual variation patterns of AOD. The

Northern Territory showed an apparent inter-annual variability of AOD compared to other states. The AODs in Tasmania

and Western Australia showed no obvious long-term tendencies and interannual variations with low values (~0.03; ~0.05)

in comparison with that in other states. The highest mean AOD (~0.055) over Australia was observed in the year of 2002.

This was largely attributed to the dust storms in southern and eastern Australia (Mitchell et al., 2010; Chan et al., 2005)

and the biomass burning in southeastern Australia (Paton-Walsh et al., 2004), which increased the aerosol loadings over

Australia. The inter-annual variation was small with mean AOD about 0.04 during the period 2003-2008. In particular,



high AOD values were observed in Australian Capital Territory (0.152) and Victoria (0.115) in 2003. This was mostly

caused by 2003 wildfires in southeastern Australia, which generated large amounts of smoke aerosols (Mitchell et al.,

2006). In 2009, there were extreme dust storms in central Australia (Mukkavilli et al., 2019), which lead to a large increase

in AOD from 0.038 (2008) to 0.045 (2009). The AOD decreased after 2009 with a mean value about 0.036 during the

period 2010-2016, and then increased from 2017 to 2020. The increasing trend in AOD during the period 2017-2020 may

be related to the more frequent wildfire activities during this period in Australia.

**3.1.2 Seasonal variations of aerosol optical properties**

Seasonal variations of AOD and AE at nine sites in Australia are presented in Fig.6. It is clear that there was a

seasonal cycle in AOD at nine sites with high values in spring and summer, and low values in fall and winter. Similar

results were reported by Mitchell et al. (2013). Further, the highest seasonal average AOD values were observed in spring

at Birdsville (0.09), Jabiru (0.22), Lake Argyle (0.23), and Lucinda (0.13), while they were observed in summer at the

other five sites (0.07-0.11). The seasonal variations in AOD observed by AERONET at nine sites were similar to that

observed by MODIS at the corresponding sites (Fig.7). Similarly, Mitchell et al. (2013) found that the AOD values at

Wagga and Canberra peaked in summer, while AOD values peaked in spring at sites that are located in the arid zone.

This is due to the increasingly forested and bushfire-prone characteristics at the more easterly sites (Mitchell et al., 2013).

However, the seasonal variation in AE was different from that in AOD. There were no obvious seasonal variations in AE

at the nine sites. The maximum seasonal mean AE values (0.92-1.43) were observed in spring at all sites except for

Canberra. Further, the seasonal mean AE was greater than 1.0 over all seasons at Canberra, Jabiru, Lake Argyle, Fowlers

Gap, and Birdsville, while the mean AE values were less than 1 over all seasons at the Adelaide Site 7 and Luncinda. In

addition, at Learmonth and Lake Lefroy, high AE values (0.98-1.07) were observed in spring and fall, and low values

(0.56-0.99) were observed in summer and winter.

Seasonal distributions of MODIS AOD are shown in Fig. 7. The spatial distribution of MODIS AOD in each season

was similar to the annual-averaged spatial distribution pattern. High AODs were observed in Spring (~0.048) and Summer

(~0.058), while low AODs were observed in fall (~0.031) and winter (~0.029). The main contributors to the high AODs

in spring and summer were smoke emissions from biomass burning, dust storms, and marine biogenic emissions (Rotstayn

et al., 2010). The occurrence frequency and intensity of dust storm activities and wildfires decreased during fall and winter,

resulting in low AOD values over Australia. One consistent feature among the spatial distributions in AOD observed in

each season were the high AODs (with different magnitude) in the northern, central, southwestern, and southeastern

Australia.

235        Fig. 8 depicts the spatial distributions of seasonally averaged precipitation and winds. Less precipitation and higher

wind speeds during spring were observed in northern Australia (north of 18°S), which may lead to the increasing AOD

from biomass burning and long-range transport marine biogenic emissions. During summer, the decreasing trend in AOD

was significant over northwestern regions, consistent with the large increase in precipitation and decrease in wind speeds.

However, the increase in AOD in eastern and southeastern regions during summer could be associated with the increase

of biomass burning and sea salt aerosols that were transported from the Pacific Ocean.

### 3.1.3 Monthly variations of aerosol optical properties

Fig. 9 depicts the monthly variations of total, fine-mode and coarse-mode AODs at 500 nm at nine AERONET sites

in Australia. Furthermore, the spatial distributions of monthly averaged MODIS AODs are shown in Fig. 10. The mean

MODIS AOD varied monthly with high values appeared during October-January and low values appeared during June-

July. In addition, significant differences in monthly AOD variations were found among regions of Australia. Therefore,

we classified the nine AERONET sites as four categories based on their locations, which are (1) Jabiru and Lake Argyle

in northern Australia, (2) Learmonth and Lake Lefory in western Australia, (3) Adelaide Site 7, Birdsville, and Fowlers

Gap in central Australia, and (4) Canberra and Lucinda in eastern Australia. In order to avoid bisect the aerosol peak, the

timeline was adopted from July to June (rather than January to December) (Mitchell et al., 2017). In northern Australia,

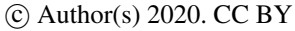

the monthly mean AODs at Jabiru and Lake Argyle varied considerably with the highest values in October (0.243; 0.264)

and the lowest values in April at Jabiru (0.066) and June at Lake Argyle (0.042). Further, fine mode AODs showed high

values during May-December at Jabiru, and during April-December at Lake Argyle. The fine mode AODs peaked in

October at Jabiru (0.172) and Lake Argyle (0.202). These results suggest that fine-mode aerosols dominated in the dry

season at the two sites, which was mostly caused by the biomass burning in northern Australia. In addition, Jabiru showed

higher mean monthly AOD values than Lake Argyle and a later increasing trend in fine mode AOD, owing to its closer

to the coast and hence to sea salt advection (Radhi et al., 2012). In western Australia, the highest monthly AOD values

occurred in January at Lake Lefroy (0.090), December at Learmonth (0.110), while the lowest monthly AOD values

occurred in June at Lake Lefroy (0.035) and August at Learmonth (0.027). Fine mode aerosols dominated in all months

at Lake Lefroy, which was mostly related to the biomass burning and anthropogenic aerosols in this region. Coarse mode

aerosols dominated in all months except for September and October at Learmonth, which was likely related to the dust

aerosols transported from the central Australia such as the Gibson Desert and Great Victoria Desert (Karlson et al., 2014).

In addition, the increasing fine mode aerosols during September-October may be a result of relatively low windspeeds in

fall at Learmonth (Fig.8), which could reduce the amount of dust that was transported from inland deserts. In the central

plains, the monthly mean AODs at Adelaide Site 7, Birdsville, and Fowlers Gap displayed two peaks. One peak was

observed in September at Adelaide Site 7 and Fowlers Gap, and in October at Birdsville, with magnitude values of 0.075,

0.068, and 0.101, respectively. The other one was observed in January with magnitude values of 0.102, 0.094, and 0.087

at Adelaide Site 7, Fowlers Gap, and Birdsville, respectively. The findings were consistent with the result of MODIS

AOD, which also showed higher values during September and February than during other months in the central plains. It

was worth mentioning that the bimodal trend weakened at Fowlers Gap and Adelaide Site 7 compared to other sites,

especially the first peak during September. This was likely due to the their farther away from the Simpson desert, which

was the most active dust source with high aerosol loading during September - February. At Birdsville, coarse mode

aerosols dominated during summer and fall, while fine mode aerosols dominated during spring and winter. Coarse mode

aerosols were more abundant during later summer and fall at Fowlers Gap, and during late spring and fall at Adelaide

Site 7. However, fine mode aerosols at Fowlers Gap and Adelaide Site 7 were more abundant than that at Birdsville due

to the influence of biomass burning aerosols transported from the eastern and southern forests aeras (Discussed in Section

3.2.2). In eastern Australia, the highest monthly AOD was observed in October (0.130) at Lucinda, and January at

Canberra (0.131). Fine mode aerosols dominated in all months at Canberra, indicating that Canberra suffered from fine

particle aerosols, such as smoke and anthropogenic aerosols. By contrast, coarse mode aerosols dominated in all months

at Lucinda, which may be related to the dust and sea salt aerosols transported from the central plains and the Pacific

Ocean, respectively.

**3.2 Classification of aerosol types and their seasonal contributions**

**3.2.1 Frequency distributions of aerosol optical properties**

The frequency distributions of AOD and AE at nine AERONET sites in Australia are shown in Fig. 11 and Fig. 12,

respectively. In general, the frequency distributions of AOD roughly showed single mode distributions, with the

maximum occurrence frequency at relatively low values (AOD < 0.1). The frequency distribution of AOD was generally

broad with high frequencies of AOD (60-94%) at the range of 0.02-0.14. Further, the frequencies of AOD with values

larger than 0.14 were generally small (4.39-18.52%) and showed a decreasing trend with AOD at all sites except for

Jabiru and Lake Argyle, where the frequencies of AOD with values larger than 0.14 were about 39.78% and 31.14%,

respectively. Moreover, at Jabiru and Lake Argyle, the frequencies of AOD with values greater than 0.30 are about 8.77 %

and 8.45%, which was mostly related to the wildfires as evidenced earlier. The histograms of AE at nine sites show that

the AE values mainly lie in the range of 0.2-1.8, which indicated the great variability of aerosol types such as biomass

burning, dust, marine and mixtures over Australia. The frequency distributions of AE were skewed towards large AE

values at all sites except for Lake Lefroy, Learmonth, and Lucinda, where AE followed the normal probability

distributions. High frequencies of AE (54.44-67.09%) with values less than 1.0 were found at Adelaide Site 7, Learmonth,

and Lucinda, suggesting the dominance of coarse mode aerosol particles. However, relatively high frequencies of AE

(52.94-79.78%) were observed at the range of 1.0 - 2.0 at the other sites, indicating a mixture of fine mode and coarse

mode aerosols at these sites.

### 3.2.2 Aerosol classification and potential source analysis

Variances in the relationship between AOD and AE provide a potential method to classify the aerosol types in

different seasons (Kaskaoutis et al., 2007). Fig. 13 depicts the scatter plots of AOD against AE in four seasons at nine

AERONET sites in Australia. It should be noted that regional transport is an important factor that can affect aerosols'

type. Therefore, the back trajectories at nine AERONET sites are simulated to analyze the potential sources of aerosol at

these sites (Fig. 14). It was clear that the aerosol concentration was high in spring and summer and low in fall and winter.

There were a wide range of AOD values ranging from 0.1 to 0.8, and AE values ranging from ~0.5 to~1.5 in all seasons

at nine sites, which indicated the existence of the mixed aerosol type. In northern Australia, Jabiru and Lake argyle

exhibited a wide range of AOD values and high AE values (>1.5) during spring, summer and winter, which suggested the

existence of biomass burning and urban/industrial aerosols. Moreover, dust aerosols (AOD>0.15, AE<0.5) were observed

during spring and summer, while clean marine aerosol was observed during fall and winter as they are located near the

Timor Sea. The back trajectories ending at Jabiru and Lake argyle showed that more than 23% of all trajectories were

from the Lake Eyre Basin (LEB), which was a significant dust source region. Further, there were about 43.40% and 21.96%

airflows originating from the northern ocean at Jabiru and Lake argyle, respectively. These results indicated that the two

sites were affected by the marine and dust aerosols, which were transported from the northwestern Indian Ocean and

southern deserts, respectively. In western Australia, it was evident that mixed aerosols, biomass burning, and

urban/industrial aerosols were dominant during all seasons at Lake Lefory. The mixed aerosols and dust aerosols were

abundant during all seasons at Learmonth, while the biomass burning and urban/industrial aerosols were also observed in



spring. The trajectories ending at Lake Lefory showed that biomass burning and urban/industrial aerosols could be transported by the western and southwestern airflows. The back trajectories ending at Learmonth showed 22.76% of airflow from the eastern deserts and 32.47% of airflow from southern inland. These results indicated that dust in eastern deserts (e.g. Gibson Desert Great Victoria Desert and Lake Eyre Basin) and urban/industrial aerosols from southern cities

(e.g. Perth) could be transported to the Learmonth. There were 44.77% of airflow from the Indian Ocean. However, clean marine aerosol was seldom found at Learmonth due to the fact that it is situated on the North-Western dust pathway (Strong et al., 2011). In central Australia, the mixed aerosols, biomass burning, and urban/industrial aerosols are found during all seasons. Dust aerosols with high AOD (>0.15) and low AE (<0.5) were observed at Birdsville and Fowlers Gap during spring and summer, while clean marine aerosols were observed during fall and spring. Different aerosol types can

be found under relatively clean atmospheric conditions with AOD <0.2 and AE < 1.5 at Adelaide Site 7. The back trajectories ending at Birdsville showed that the southern (25.42%) and southwestern (19.28%) airflows may bring the clean marine aerosols and dust aerosols from the Indian Ocean and Lake Eyre Basin to Birdsville, while the eastern (25.42%) and southeastern (33.31%) airflows may bring the biomass burning aerosols to Birdsville. The back trajectories ending at Fowlers Gap and Adelaide Site 7 showed that more than 67% of airflow were originated mainly from the Indian

Ocean, which could transport clean marine aerosols to the two sites. Further, there were 32.71% and 29.53% of airflows at Fowlers Gap and Adelaide Site 7 from the southeastern Australia, which implies a possible transport of biomass burning aerosols. In eastern Australia, the Canberra site exhibited a wide range of AOD values and high AE values ranging from ~1.5 to~2.5, which indicated the existence of biomass burning and urban/industrial aerosols during all seasons. In addition, clean marine and dust aerosols were observed during fall and winter. The back trajectories ending at Canberra

showed 54.03% of airflow from southwest and 45.97% of airflow from southeast, suggesting a possible transport of biomass burning and clean marine aerosols from forest regions and ocean. Different aerosol types (e.g. clean marine aerosols, dust, mixed aerosols) were observed at Lucinda during all seasons. The back trajectories ending at Lucinda



illustrated that the airflows from southwest and northwest may bring dust and biomass burning to Lucinda, respectively.

As indicated earlier, the relationship between AOD and AE can be used to distinguish aerosol types. However, it is

difficult to discriminate biomass burning from urban/industrial using this relationship alone (Mishra and Shibata, 2012).

Further, a better analysis can be made through correlations between SSA and AE. Fig. 15 represents the relative

percentages of different aerosol components in each season at nine sites based on aerosol classification method of Giles

et al. (2012). It is clear that the mixed type of aerosol dominated in all seasons at nine sites. Moreover, the mixed type of

aerosols was mainly a mixture of biomass burning and dust aerosols. Mitchell et al. (2017) also found that the aerosol

sources over Australia were driven by mechanisms that do not vary greatly either on regional or continental scales by

analyzing the monthly mean data at 22 sites. They pointed out this is associated with the intercontinental transport of

biomass burning aerosol. In northern Australia, biomass burning aerosols accounted for a relatively large proportion at

Jabiru and Lake Argyle in spring, autumn and winter, which was related to the biomass combustion during the dry season.

In addition, dust aerosols were also observed during summer, autumn and winter. Fire events were also a main provider

of dust aerosols in the atmosphere because the pyro-convection could accelerate the dust entrainment during the fire

events. The results indicated that the aerosols at the two sites were affected by the fire-related dust emissions and dust

transported from the southeastern deserts. In western Australia, the urban/industrial aerosols were observed during all

seasons, making fine mode aerosols dominated throughout the year (Section 3.1.3) at Lake Lefory. The urban/industrial

aerosols over Lake Lefory may originate from Perth, which were transported to Lake Lefory by the western airflow. The

Learmonth was found with heavy loadings of dust aerosols, which is consistent with our earlier result. In eastern Australia,

dust aerosol was another dominant aerosol type at Lucinda. The classification results may be inaccurate due to the lack

of SSA data at Canberra. However, many previous studies showed that urban/industrial aerosols and biomass burning

aerosol were the main components of aerosols at Canberra (Mitchell et al., 2006; Provençal et al., 2017).





### 3.2.3 Spatio-temporal characteristics of different aerosol types

MERRA-2 data was used to determine the contribution of different kinds of aerosols to the AOD over Australia.

Considering the similar emission sources of Organic Carbon and Black Carbon aerosols, we combined the two as

carbonaceous aerosol for analysis. Carbonaceous aerosol and sulfate aerosol are produced from biomass burning, fossil

fuel combustion, biofuel consumption, while dust and sea salt aerosols mainly originate from natural emissions (Randles

et al., 2017). The spatial distributions of carbonaceous, dust, sulfate, sea salt AOD over Australian continent are shown

in Fig. 16. Carbonaceous aerosols were mainly distributed in northern and southeastern Australia. Carbonaceous aerosols

in these two regions could be highly related to the fires in the grasslands, forests, and croplands during the dry seasons.

Dust aerosols were mainly distributed in the central plains of Australia. The dust aerosols over the central plains primarily

originated from the Lake Eyre Basin, one of the southern hemisphere's most significant dust sources. Mukkavilli et al.

(2019) found that the spatial distributions of dust aerosols across Australia demonstrated concentrated values in the Lake

Eyre Basin by using the ECMWF Monitoring Atmospheric Composition and Climate (MACC) reanalysis product. Sulfate

aerosols were mainly observed in the northwestern (such as Darwin) and southeastern (such as Melbourne, Canberra, and

Sydney) Australian coastal regions, where human activities were highly frequent. The near coastal region generally had

higher sea salt aerosol loadings than the continental interior region due to the land-sea breeze effects. Prijith et al. (2014)

found that the higher wind speed would lead to more sea salt aerosol formation, and the corresponding shorter transport

time would lead to weaker loss. Sea salt aerosols had a relatively large impact on the northern coastal regions of Australia,

which were mostly due to the higher wind speeds in the northern coast (Fig.8). Overall, carbonaceous over northern

Australia, dust over central Australia, sulfate over densely populated northwestern and southeastern Australia, and sea

salt over Australian coastal regions were the major types of atmospheric aerosols in Australia.

       To determine the temporal distributions of different kinds of aerosols in Australia, seasonal variation analyses of

aerosols are performed. Figure 17 shows the seasonal variations of carbonaceous, dust, sulfate, and sea salt aerosols in



Australia. Generally, carbonaceous AODs in northern regions were much higher than that in southern and southwestern

regions in all seasons. High carbonaceous AOD values were observed in northern Australia during spring and southeastern

Australia during summer. This was consistent with our findings in seasonal distributions of MODIS AOD in Australia.

Further, both northern and eastern Australian atmosphere were influenced by carbonaceous aerosols in spring. During

spring and summer, the northern and southern Australia experienced warm, dry, and high wind speeds weather conditions,

which increased the occurrence frequency and development of extreme wildfires. The carbonaceous AOD declined in fall

and winter in Australia due to the decreasing occurrence frequency of biomass burning. The central Australia had always

been the relatively high dust AOD center in four seasons. Further, higher dust AOD values were observed in spring and

summer and lower dust AOD values were observed in fall and winter. This seasonality of dust aerosols was consistent

with the results of Mukkavilli et al. (2019) and Ridley et al. (2016). The near coastal regions of Australia generally had

higher wind speeds and precipitation than the continental interior regions (Fig. 8). Therefore, these regions had lower dust

storm activities than the continental interior regions. In contrast, the west, the central, and parts of the south Australian

continents had dry spring and summer, which enhanced the dust storm occurrence frequency. The frequencies of dust

storms in central Australia showed strong seasonal dependency on the frontal activity (i.e. pre-frontal northerly, frontal

westerly and post-frontal southerly winds) (Baddock et al., 2015; Strong et al., 2011). During the austral winter, high

pressure systems were situated in the middle of the continent and blocked the fronts from reaching the continent, which

helped the formation of stable weather conditions with little dust entrainment. During late spring and early summer, the

high pressure systems shifted southwards with frontogenesis arising, which increased the occurring chances of dust storms

(Strong et al., 2011; Ekström et al., 2004). High sulfate AODs were observed in densely populated coastal urban regions

in southeast and northwest Australia during spring and summer, while low sulfate AODs were observed during fall and

winter. The high sulfate AODs during spring and summer could be a result of the high local emissions from industrial

activities and biomass burning, along with the adverse meteorological conditions such as low wind speeds and weak

precipitation. Sea salt AODs showed a decreasing trend from the coast to the further inland regions in all seasons. High

sea salt AOD values were observed in northeast coastal regions. Further, high sea salt AOD values appeared in summer

and fall, while low sea salt AOD values appeared in spring and winter. Actually, many studies have identified the large

increase of coarse mode sea-salt aerosols at coastal regions of northern Australia during the wet season (i.e. November to

April) (Bouya et al., 2010; Radhi et al., 2012).

**3.3 Vertical distribution of aerosol optical properties**

The CALIPSO aerosol product can provide the profiles of aerosol optical properties. Thus, three domains are chosen

to explore the vertical distribution of various aerosol types in Australia, as shown in Fig. 18. The first domain covers most

of Australian continent (blue shade), where carbonaceous over northern Australia and dust over central Australia are the

major types of atmospheric aerosols. The second domain includes parts of Northern Territory, Western Australia, and

Queensland (deep purple shade), where biomass burning aerosol dominates. The third domain includes parts of Northern

Territory, Southern Australia, New South Wales, and Queensland (gray shade), where dust aerosol is abundant.

Fig. 19 shows the occurrence frequency profile of each aerosol type in each season from 15-year CALIPSO

observations in Australia. In general, polluted dust was the dominant aerosol type detected roughly from 0.5 to 6 km

during spring and summer, and roughly from 0.5 to 4 km during fall and winter. The altitude with the peak occurrence

frequency (>9%) of the polluted dust was ~1 km throughout the year. Burton et al. (2013) have pointed out that the

polluted dust was defined as a mixed aerosol type to represent mixtures of dust and biomass burning smoke or pollution.

However, polluted dust in Australia was closer to the mixture of biomass burning smoke and dust at most regions. These

results also confirmed that the mixed type of aerosols (mostly biomass burning and dust) is the dominant type during all

seasons in Australia as discussed in section 3.2.2. Polluted continental was the secondary dominant aerosol type detected

approximately at heights from 1 to 2 km, with higher occurrence frequency during fall and winter than that during spring





and fall. The occurrence frequency of clean marine was the largest at the surface and decreased with altitude. The

occurrence frequency of elevated smoke increased with height at 0-3 km, and then decreased with height at 3-12 km.

Elevated smoke was the secondary dominant aerosol type detected approximately at heights from 2 to 5 km. Moreover,

elevated smoke was the dominant aerosol type at 6-12 km during spring and winter, and at 6-8 km during fall and winter.

Dust with an occurrence frequency larger than 0.1% can reach as high as 12 km. Dust was detected more frequently at

height approximately from 0 to 6 km in all seasons. Further, the occurrence frequency of the dust was significantly higher

in summer than in other three seasons, which is consistent with the observations of MERRA-2.

The vertical profile of occurrence frequency for each type of aerosol in each season over the biomass burning regime

in Australia is presented in Fig. 20. Polluted dust was the dominant aerosol type at heights approximately from 0.5 to 3

km during all seasons except for summer when polluted dust dominated at heights roughly from 0.5 to 6 km. The largest

occurrence frequency of polluted dust was observed at heights from 0 to 2 km during fall. The wind fields in fall (Fig.8c)

and the back trajectory at Jabiru and Lake Argyle (Fig.15e and 15f) revealed that the airflows passed over the Lake Eyre

Basin before reaching the biomass burning regime areas. Those airflows brought a large amount of dust from the Lake

Eyre Basin, which then got mixed with the biomass burning aerosols, resulting in the increasing occurrence frequency of

polluted dust during fall. Similar to the results found for Australian continent, the altitude with peak occurrence frequency

(~5%) for elevated smoke was ~3 km throughout the year. Higher occurrence frequency of elevated smoke was observed

at heights from 2 to 4 km in spring, which could be associated with the biomass burning during dry season. Polluted

continental was the secondary dominant aerosol type at heights from 0.5 to 2 km, with higher occurrence frequency during

fall and winter than during spring and summer. In addition, the dust occurrence frequency increased with height from 0

to 4 km during summer. The finding is consistent with the results shown in Fig. 18, which showed high dust aerosol

loadings in the biomass burning regime areas during summer. The results also confirmed the existence of dust aerosols

in northern Australia, which were mostly generated along with fires and transported from south inland deserts.

Figure 21 shows the occurrence frequency of aerosol type over desert regime areas in Australia. Similar to the biomass burning regime areas, polluted dust was the dominant aerosol type at heights from 0 to 5 km with higher occurrence frequency (>10%) during fall and winter. Similar results were reported by Huang et al. (2013), who found

that the aerosol characteristics in western Australia were closer to that in the biomass burning regime rather than that in other desert regimes such as northern Africa and west China. There were two likely main reasons. First, the continent had relatively low topographical relief, and the arid regions were old and highly weathered. Thus, fine particles were blown away a long time ago (Prospero et al., 2002). Second, the atmosphere of the central Australia was affected by the biomass burning aerosols transported from eastern and southeastern Australia as discussed earlier. In addition, the elevated smoke

was the dominant type at heights above 5 km in all seasons except winter when the elevated smoke was dominant at heights above 3 km. The smoke in biomass burning regime was mostly transported from the eastern and southeastern Australia (Fig. 14). However, there are significant differences in aerosol types between biomass burning and dust regime areas. Dust was the secondary dominant aerosol type at heights from 0 to 3 km in spring, from 0 to 4 km in summer, from 0 to 2 km in fall. The altitude with peak occurrence frequency of dust was ~1 km throughout the year. Furthermore, there

were no clean marine and dusty marine aerosols in the desert regime areas due to its far away from the ocean. Although polluted dust was the dominant aerosol type at heights from 0 to 5 km over the desert regime, the occurrence frequency of dust significantly increased at the heights from 0 to 3 km. Furthermore, higher occurrence frequencies of the polluted dust were found during all seasons over the desert regime than that over the biomass burning regime.

In general, results from CALIPSO indicated that the dominant aerosol type at heights from ~0.5 to ~3 km in Australia

was the polluted dust, which is consistent with the aerosol classification results at the nine AERONET sites. This resulted from intercontinental transport of aerosols over Australia, especially for biomass burning aerosols. However, the dust-prone characteristic of aerosol was more prominent in the central Australia, while the biomass burning-prone



characteristic of aerosol was more prominent in northern Australia.

**4. Conclusions**

In this study, long-term spatiotemporal variations of aerosol optical properties in Australia were analyzed by using a combination of ground-based and satellite aerosol products. Two different methods based on the different combinations of aerosol optical properties from AERONET products were used to classify the aerosol types at nine sites. Furthermore, the spatiotemporal variations and vertical distributions of different aerosol species were analyzed by using MERRA-2 and CALIPSO data. The main conclusions of the study are as follows.

1. The annual averaged AOD at most Australia sites showed increasing trends (0.002-0.028 yr$^{-1}$) during the observation period except for Canberra, Jabiru, and Lake Argyle, at which the AOD showed decreasing trends (-0.004 - -0.002 yr$^{-1}$). By contrast, the annual averaged AE presented decreasing trends at most sites (-0.044 - -0.005 yr$^{-1}$). There was a clear seasonal variation in AOD with high values in austral spring and summer and low values in austral fall and winter. However, the seasonal variations of AE were weaker with the highest mean values in spring at most sites. Furthermore, the monthly variation of AOD tends to have a unimodal distribution with peak values in September-January at most sites, which were located in warm and rainy regions. On the contrary, the monthly variation of AOD at those sites in arid regions more tends to have a bimodal distribution, showing dual peaks in austral spring and summer.

2. During the period July 2002- May 2020, the annual average MODIS AOD showed a weak decreasing trend (-0.0003 yr$^{-1}$) in Australia. The spatial distribution of annual mean MODIS AOD showed obvious spatial heterogeneity with higher values in the east than in the west Australia. High aerosol loadings were observed in the Australian tropical savanna regions, Lake Eyre Basin, and southeastern regions of Australia, while low aerosol loadings were observed over the arid region in western Australia. High AOD values were observed in spring and summer, while low AOD values were

observed in fall and winter. The AOD increased from August until January of the following year in the northern, central,

and eastern Australia, which was most likely related to the biomass burning and dust storms.

3. The mixed type of aerosols (biomass burning and dust aerosols) was dominant in all seasons identified at nine sites. The biomass burning aerosol was the second dominant contributor to the aerosol composition at Jabiru and Lake Argyle, while the dust aerosol was the second dominant aerosol type at Birdsville, Fowlers Gap, Adelaide Site 7, Lucinda, and Learmonth. The clean marine and urban/ industrial aerosols were observed at most sites such as Lake Lefory and Canberra.

Spatially, aerosols in Australia were mainly carbonaceous over the northern regions, dust over the central regions, sulfate over densely populated northwestern and southeastern regions, and sea salt over coastal regions. Seasonal variations of carbonaceous, dust, sulfate, and sea salt AODs in Australia generally showed high values in spring and summer and low values in fall and winter. Furthermore, the results from CALIPSO showed that clean marine and polluted dust were the dominant aerosol type at heights approximately from 0 to 5 km in all seasons, while polluted continental was the secondary

dominant aerosol type at heights from 1 to 2 km, and elevated smoke was the secondary dominant aerosol type at heights from 2 to 5 km in Australia. Australian aerosol tends to have the similar source characteristics due to intercontinental transport of aerosols in Australia, especially for biomass burning and dust aerosols. However, the dust-prone characteristic of aerosol was more prominent in the central Australia, while the biomass burning-prone characteristic of aerosol was more prominent in northern Australia.

The results of this study provide significant information on aerosol optical properties in Australia. Of course, there are still several limitations. First, only nine AERONET sites were used, which may not be sufficient to fully reveal aerosol optical properties in Australia. Thus, in the future, more in-depth analyses should be made using long-term ground-based measurements at more sites and short-term field campaigns in conjunction with satellite remote sensing. Second, only two aerosol classification methods were used to distinguish the aerosol types at nine AERONET sites, which could pose

a potential threat to the accuracy of the classification results of aerosol types. Hence, more aerosol classification methods

and aerosol optical parameters (e.g. fine mode fraction (FMF), Extinction Angstrom Exponent (EAE), Abortion Angstrom Exponent (AAE), Real and Imaginary index of Refraction (RRI & IRI respectively), Asymmetry Parameter (ASY)) could be used to determine the aerosol types in Australia.

**Acknowledgements.**

This research was supported by the National Natural Science Foundation of China (Grants 41925022), the China National Key R&D Program (2019YFA0606803), and the State Key Laboratory of Earth Surface Processes and Resource Ecology. The authors would like to thank CALIPSO team, NASA AERONET Network, NASA Goddard Space Flight Center (GSFC), and NASA Global Modeling and Assimilation Office (GMAO) for providing aerosol optical properties data. We

also thank the European Centre for Medium-Range Weather Forecasts team for processing and distributing the ERA-5 data.

**Data Availability.**

The Australian DEM are provided by Australian public data website (https://data.gov.au/data/). The Dynamic Land Cover Dataset (DLCD) are provided by Geoscience Australia (http://www.ga.gov.au/scientific-topics/earth-

obs/accessing-satellite-imagery/landcover). The AERONET dataset can be found at https://aeronet.gsfc.nasa.gov/. Aqua MODIS AOD data were retrieved from the Atmosphere Archive and Distribution System Distributed Active Archive Center (LAADS DAAC; https://ladsweb.modaps.eosdis.nasa.gov/). HYSPLIT data are accessible through the NOAA READY website (http://www.ready.noaa.gov). MERRA-2 Reanalysis data were provided by the NASA Global Modeling and Assimilation Office (https://gmao.gsfc.nasa.gov/reanalysis/MERRA-2/). ERA-5 Reanalysis

data were provided by the European Centre for Medium Weather Forecasts (https://cds.climate.copernicus.eu/). The CALIPSO data were downloaded from the NASA's website (https://www-calipso.larc.nasa.gov/).



**Author contributions.**

CFZ and XCY developed the ideas and designed the study. YKY contributed to collect and analyses CALIPSO aerosol data. XCY performed the analysis and prepared the manuscript.

**Competing interests.**

The authors declare that they have no conflict of interest.

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

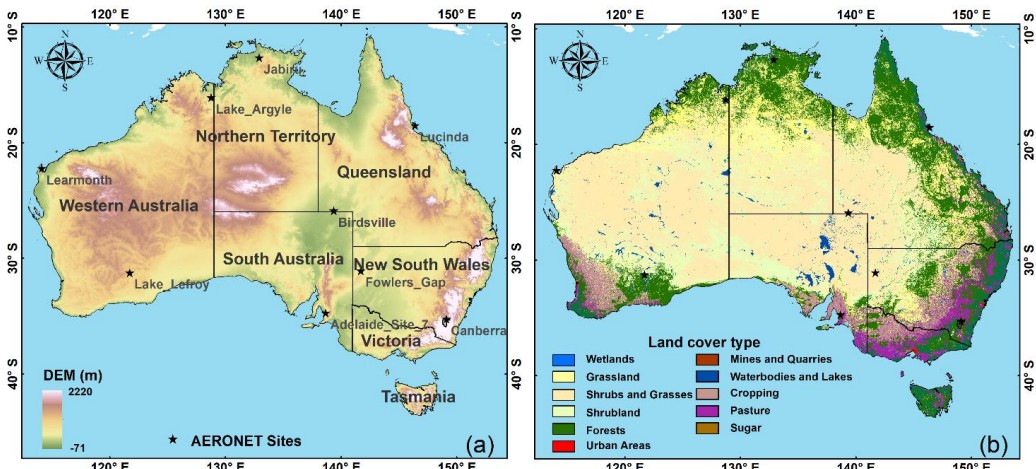


**Figure 1. Terrain elevation of Australia and the nine AERONET sites used are indicated by black star (a). Land cover type in**

**2015 of Australia by using the Dynamic Land Cover Dataset (DLCD) Version 2.1 dataset (b). Note: The Australian DEM Data**

**are provided by Australian public data website (https://data.gov.au/data/). The Dynamic Land Cover Dataset (DLCD) are**

**provided by Geoscience Australia (http://www.ga.gov.au/scientific-topics/earth-obs/accessing-satellite-imagery/landcover).**





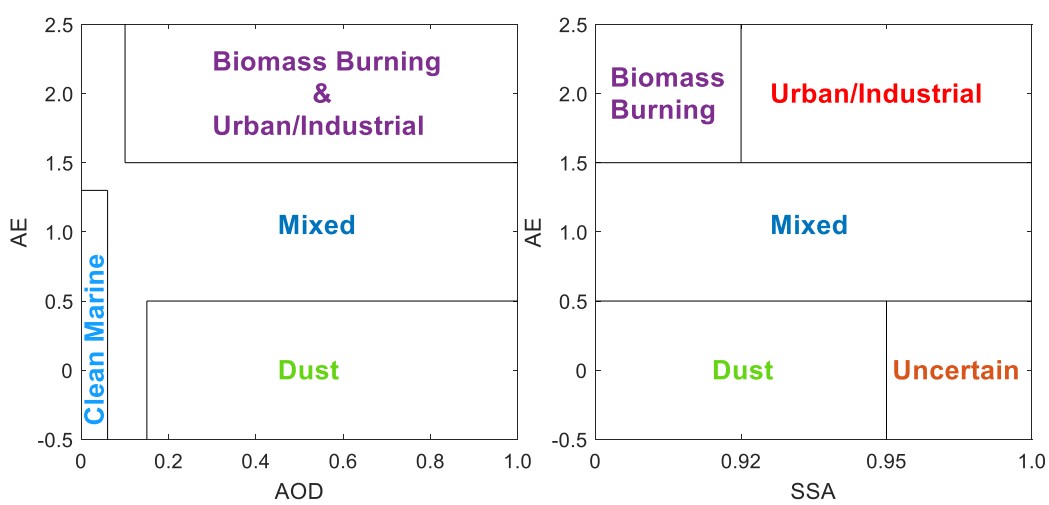


**Figure 2. Schematic diagrams of the aerosol classifications defined in Kaskaoutis et al. (2007) (Left) and Giles et al. (2012)**

**(Right).**

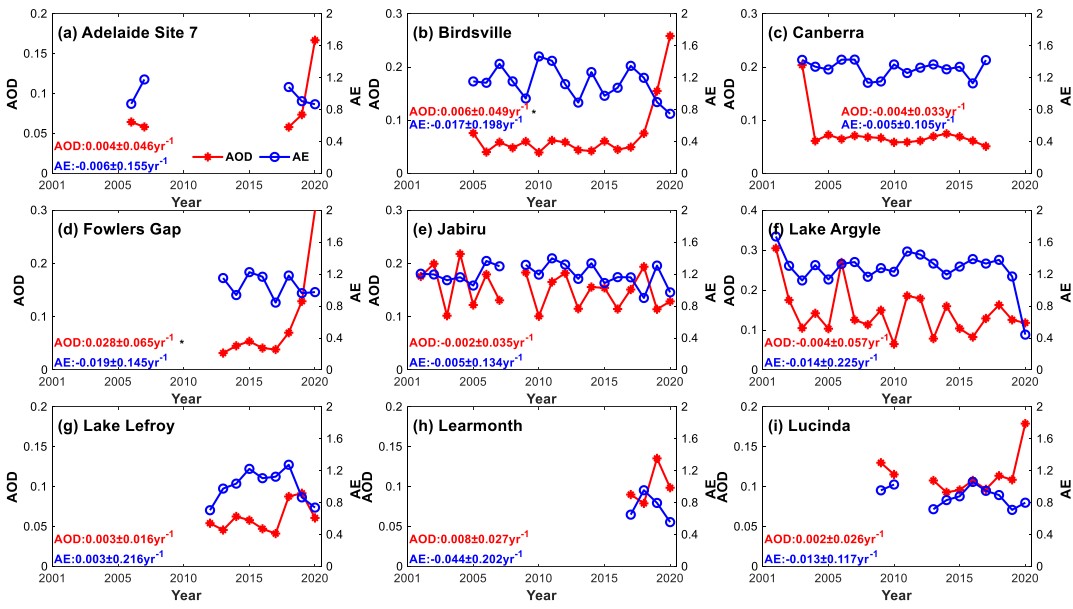

**Figure 3. Temporal variations of annual mean AOD, AE at nine AERONET sites in Australia. Note: "*" means passing the**

**confidence testing at α=0.05.**



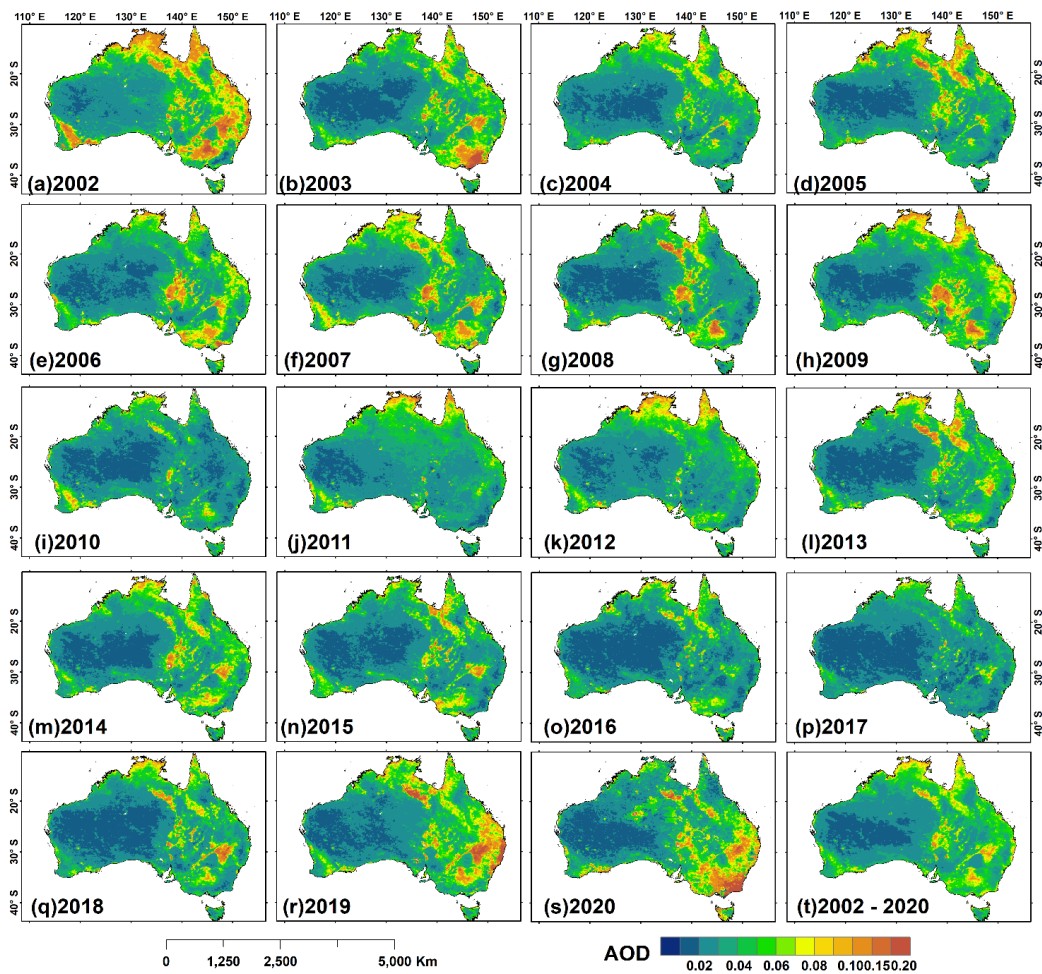

**Figure 4. Spatial distributions of annual averaged Aqua MODIS DB AODs at 550 nm in Australia from 2002 to 2020 (a-s),**

**along with that for multi-year averaged Aqua MODIS DB AODs at 550 nm in Australia from 2002 to 2020 (t). Note: the year**

**of 2002 only covers July to December and the year of 2020 only covers January to May.**




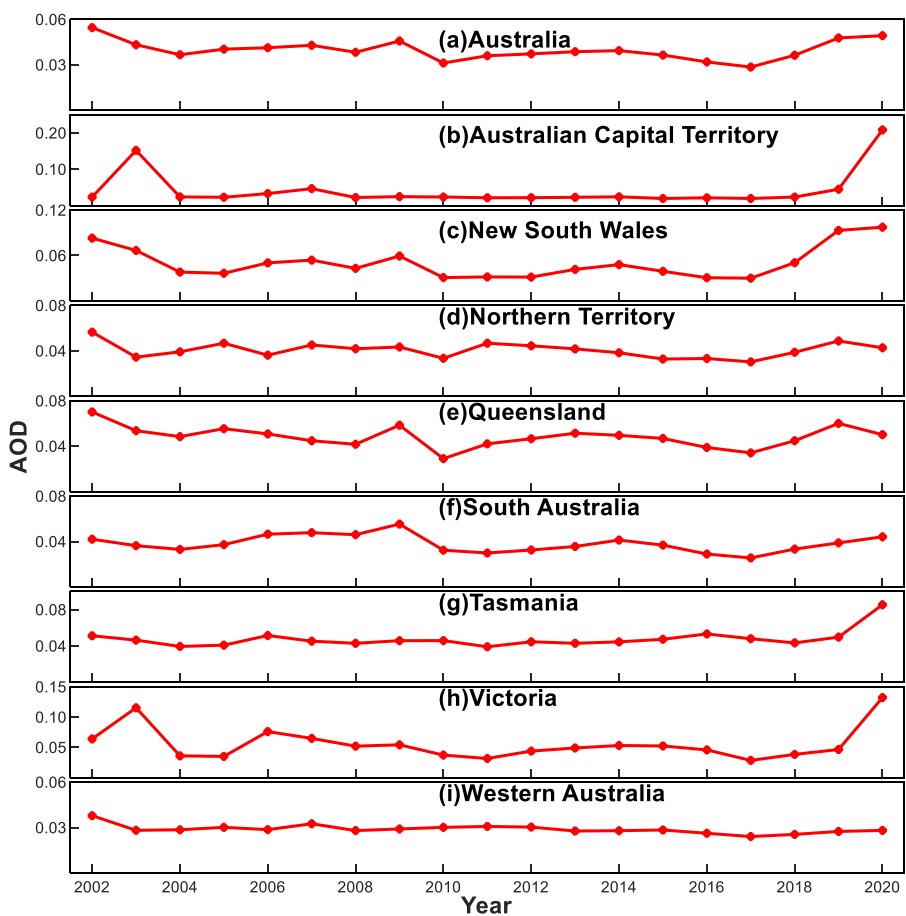

**Figure 5. Time series of the yearly averaged Aqua MODIS DB AODs over Australia and its 8 main administrative regions**

**from 2002 to 2020.**



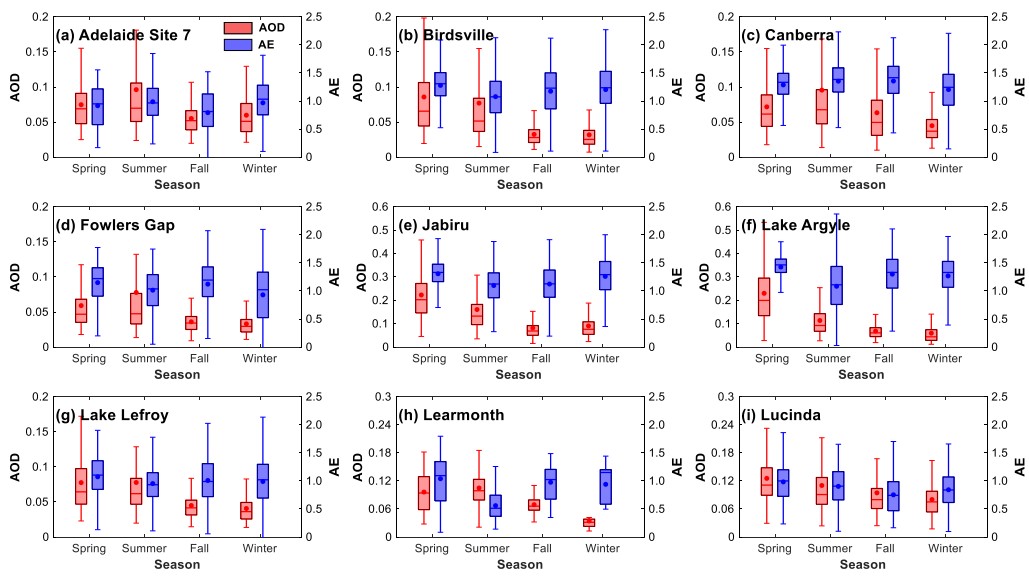

**Figure 6. Box plots of the seasonal average AOD and AE at the nine AERONET sites in Australia. In each box, the dots in the center are the mean, and the lower and upper limits are the first and the third quartiles, respectively. The lines extending vertically from the box indicate the spread of the distribution with the length being 1.5 times the difference between the first and the third quartiles.**

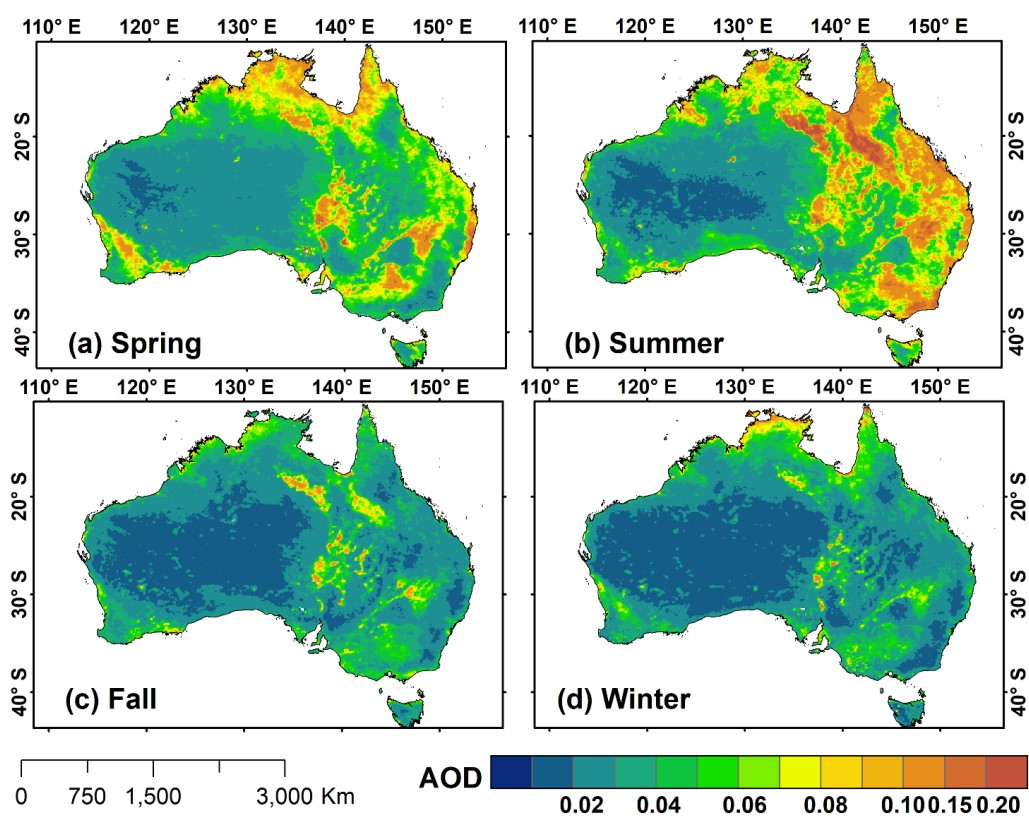


**Figure 7.** Spatial distributions of seasonally averaged Aqua MODIS DB AODs at 550 nm during the period 2002-2020 in

Australia, a) spring, b) summer, c) fall, and d) winter.

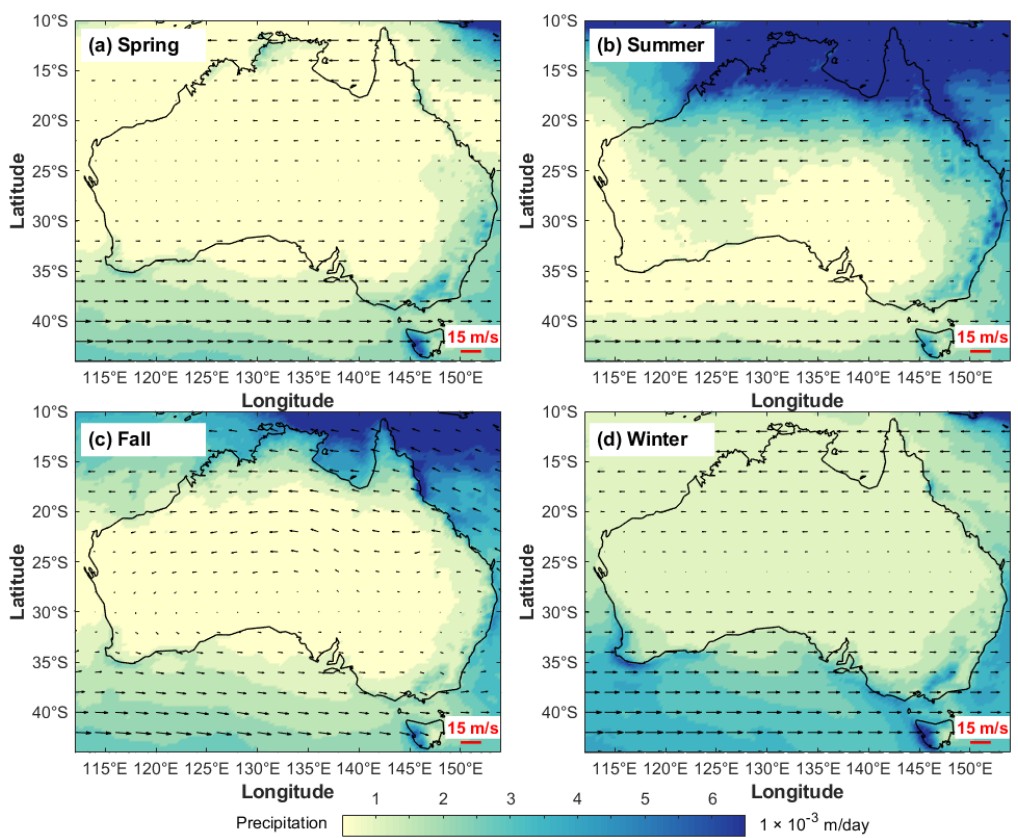

**Figure 8. Spatial distributions of seasonally averaged total precipitation (colors) and winds (arrows) from the ERA-5 monthly**

**dataset in Australia, a) spring, b) summer, c) fall, and d) winter.**



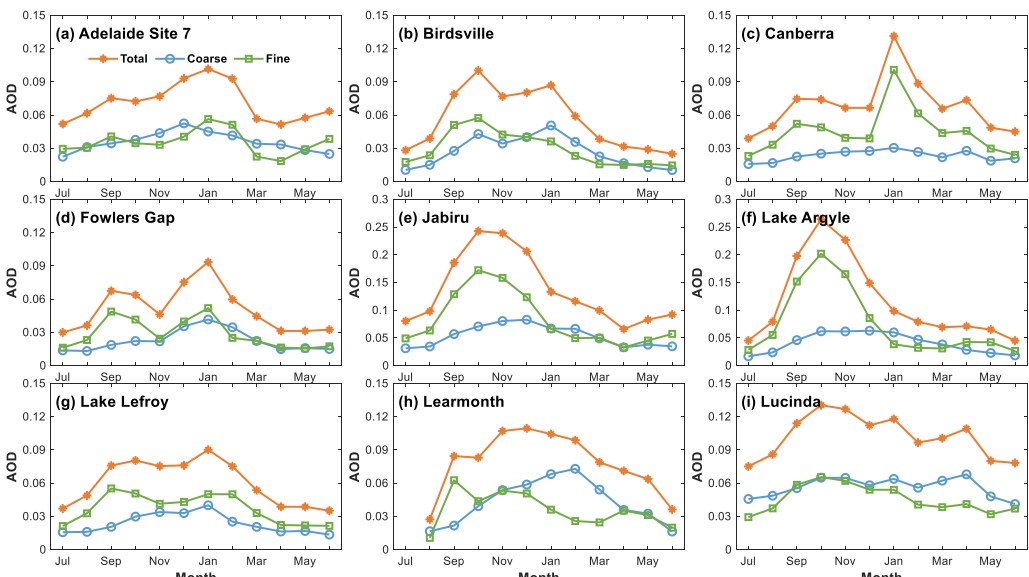

**Figure 9. Monthly variations of multi-year averaged total, fine-mode, and coarse-mode AODs at 500 nm at nine AERONET**

**sites in Australia.**

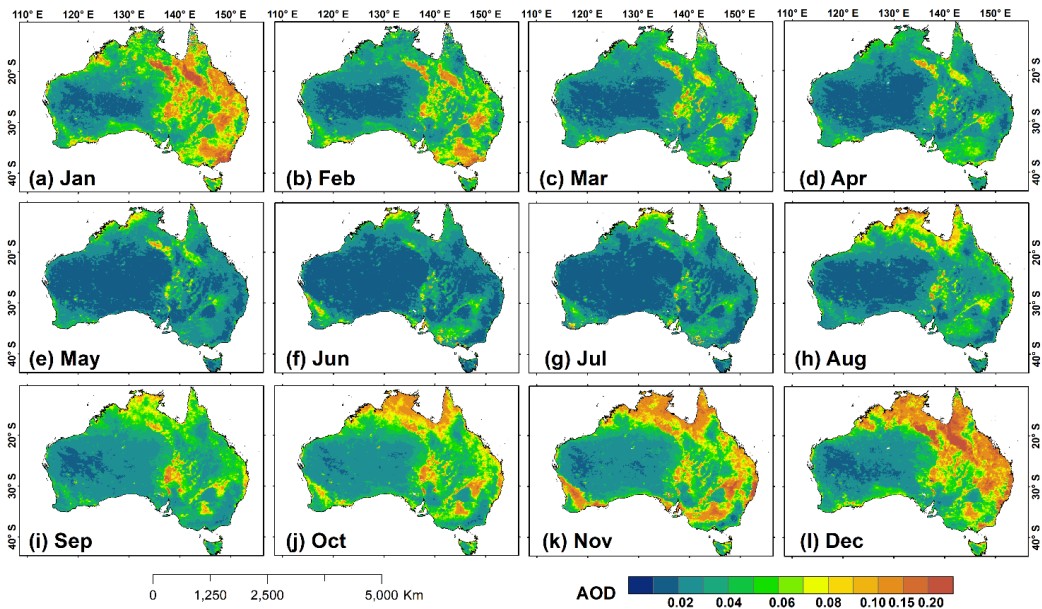

**Figure 10. Spatial distributions of monthly averaged Aqua MODIS DB AODs at 500 nm during the period 2002-2020 in**

**Australia.**





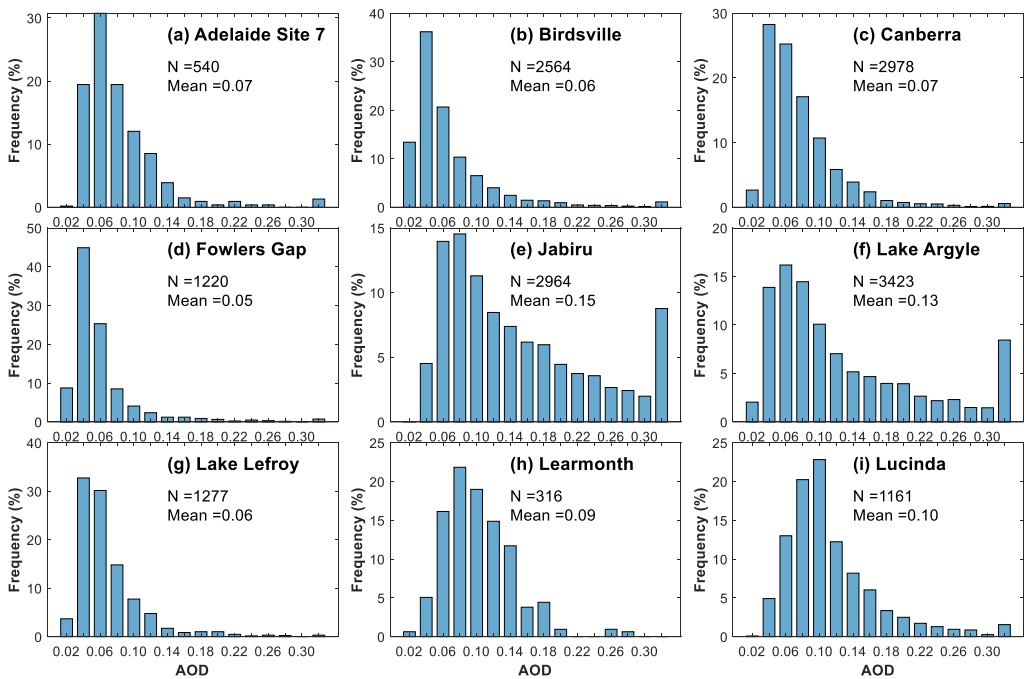

**Figure 11. Occurrence frequencies of AODs at nine AERONET sites in Australia.**

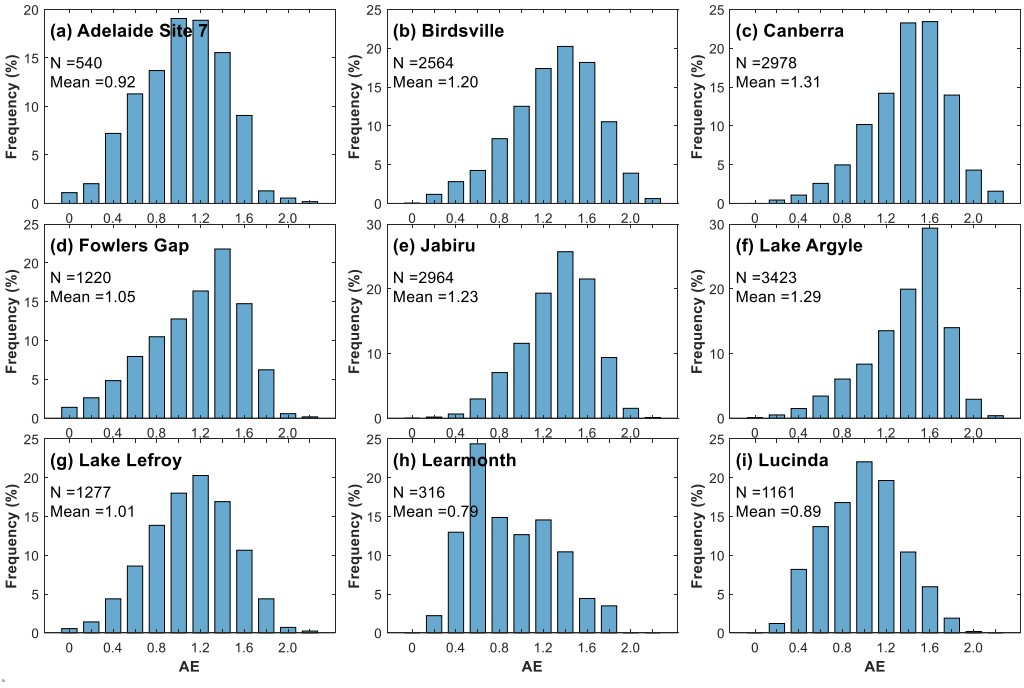

full



**Figure 12. Occurrence frequencies of AE at nine AERONET sites in Australia.**

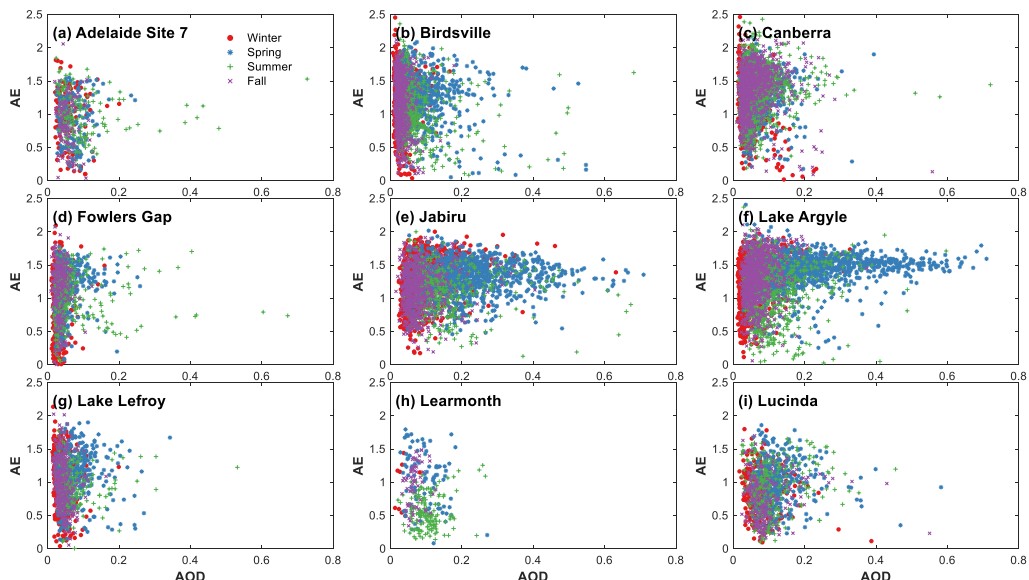

**Figure 13. Scattered plot for AOD at 500 nm vs AE at 500 nm in spring (blue), summer (green), fall (purple), winter (red) at**

**nine AERONET sites in Australia.**

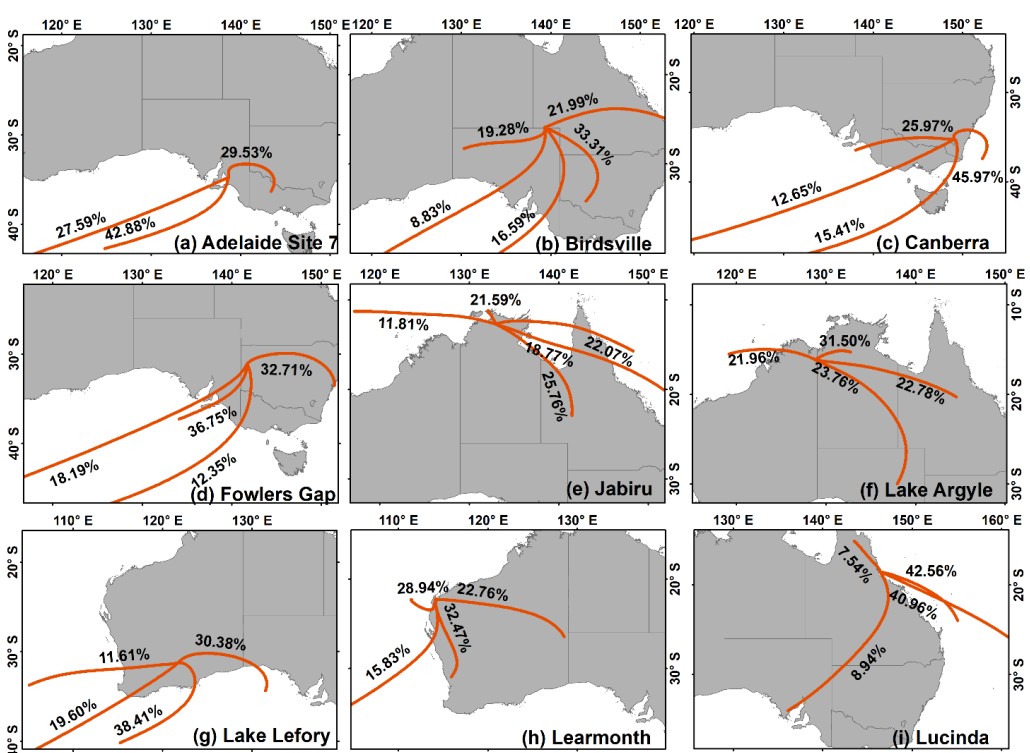

**Figure 14.** Cluster analysis of simulated back trajectories from HYSPLIT during the period January 2005-May 2020 for air

masses ending at nine AERONET sites at 500 m above ground level in Australia.





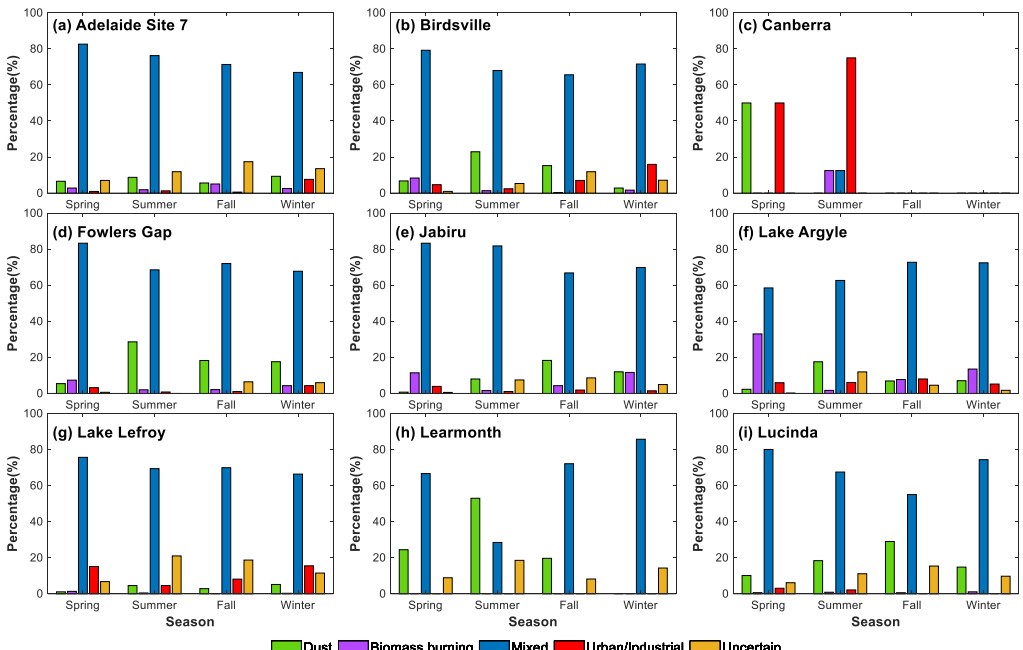

**Figure 15. Relative percentages of different aerosol components in each season at nine sites during the observation period,**

**including dust, biomass burning, mixed, urban/industrial, and uncertain aerosol types.**



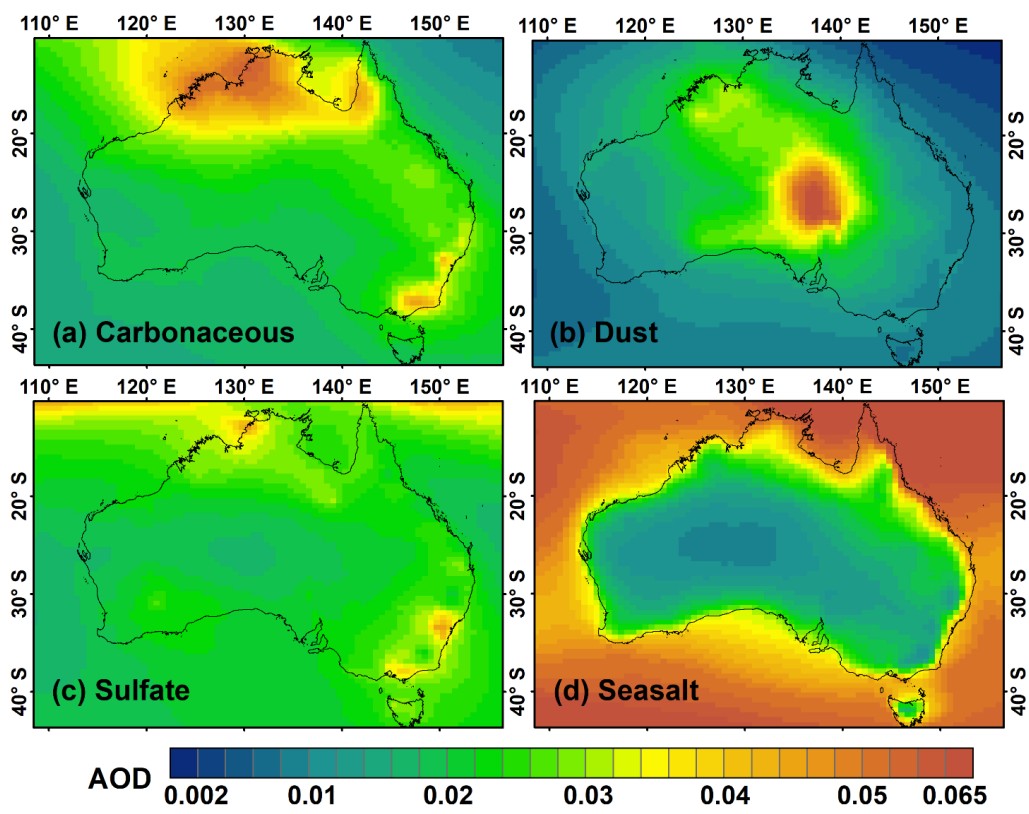


**Figure 16. Spatial distributions of carbonaceous (a), dust (b), sulfate (c), and sea salt (d) AOD derived from MERRA-2**

**monthly dataset during the period 2002-2020 in Australia.**

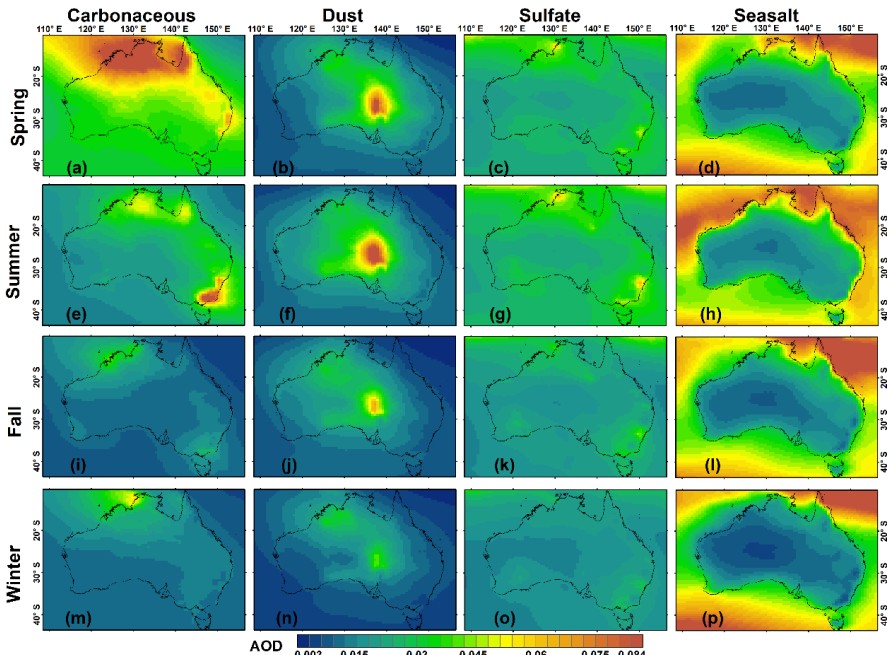

**Figure 17. Spatial distributions of carbonaceous, dust, sulfate, and sea salt AOD derived from MERRA-2 monthly dataset in**

**each season during the period 2002-2020 in Australia.**





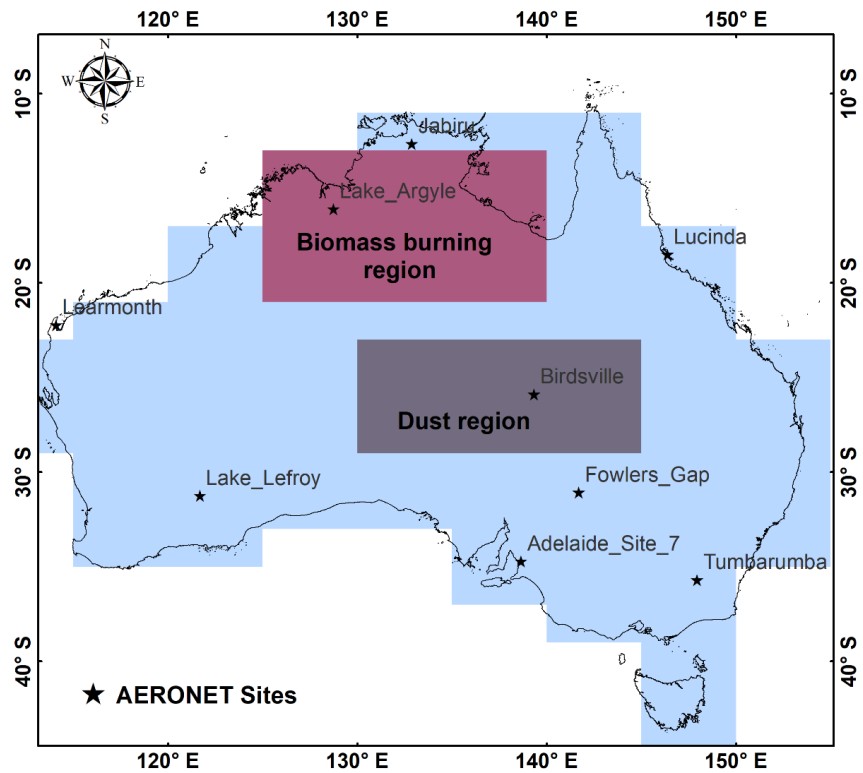

**Figure 18. Three domains selected for vertical profile analysis in this study. Blue shade indicates the Australia; Deep purple**

**shade represents biomass burning regime areas; and Gray shade indicates desert regime areas.**



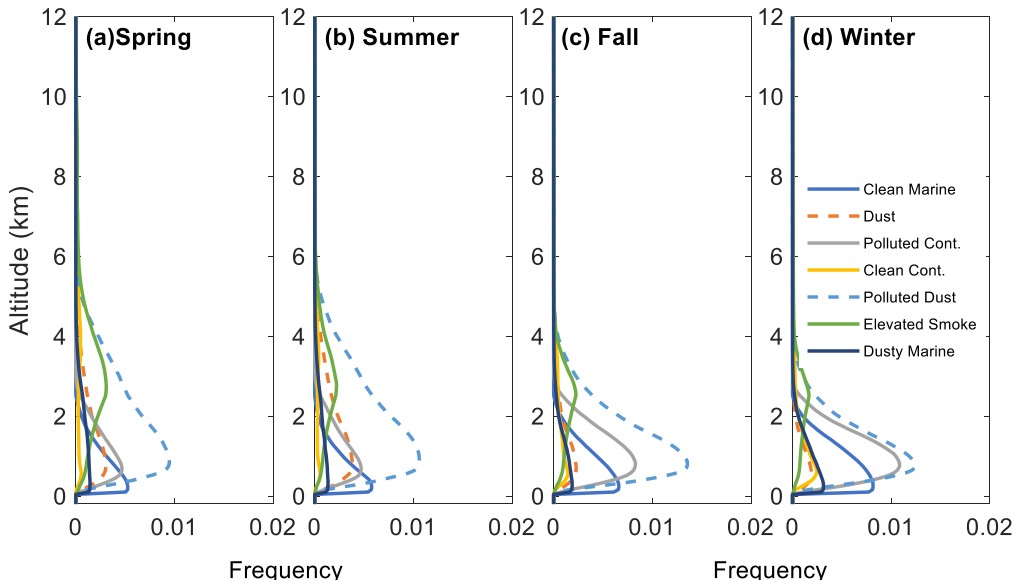

**Figure 19. Occurrence frequency profile of each aerosol type in four seasons from 15-year CALIPSO L3 aerosol profile data**

**product in Australia: a) spring, b) summer, c) fall, and d) winter.**

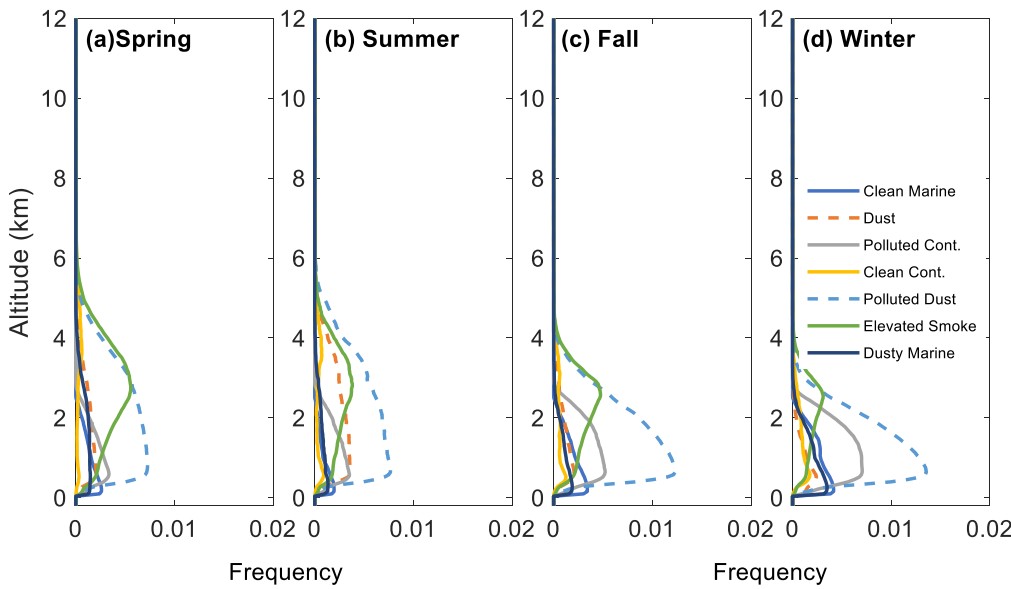





**Figure 20. Occurrence frequency profile of each aerosol type in four seasons from 15-year CALIPSO L3 aerosol profile data product in biomass burning regime in Australia: a) spring, b) summer, c) fall, and d) winter.**

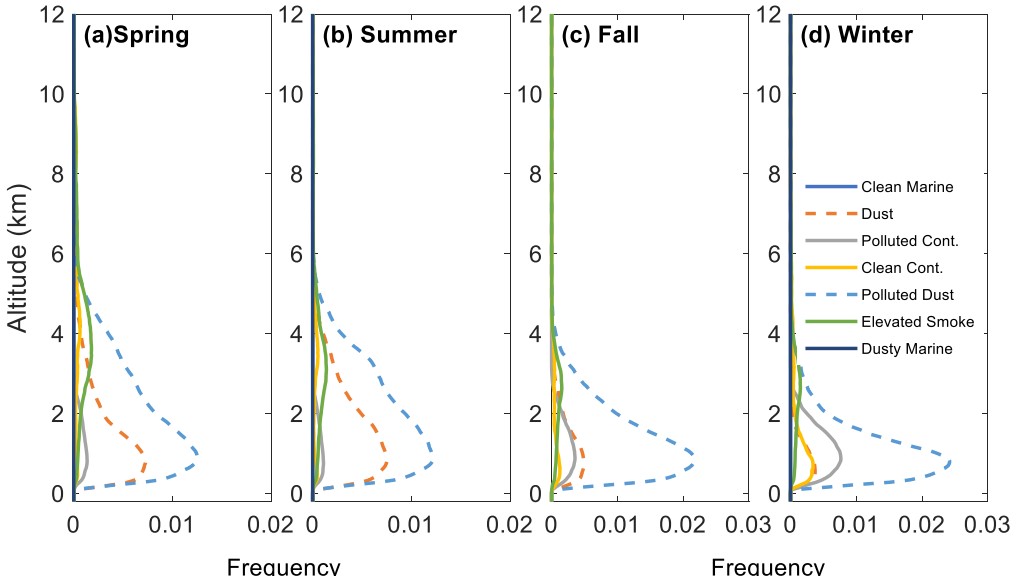


**Figure 21. Occurrence frequency profile of each aerosol type in four seasons from 15-year CALIPSO L3 aerosol profile data product in desert regime in Australia: a) spring, b) summer, c) fall, and d) winter.**







**Table 1. Site location and data time period at each site.**

| Site | LON | LAT | Time span |
|------|-----|-----|-----------|
| Adelaide_Site_7 | 138.66 | -34.73 | 2006-2007;2017-2020 |
| Birdsville | 139.35 | -25.90 | 2005-2020 |
| Canberra | 149.11 | - 35.27 | 2003-2017 |
| Fowlers_Gap | 141.70 | -31.09 | 2013-2020 |
| Jabiru | 132.89 | -12.66 | 2001-2007;2009-2020 |
| Lake_Argyle | 128.75 | -16.11 | 2001-2020 |
| Lake_Lefroy | 121.71 | -31.26 | 2012-2020 |
| Learmonth | 114.10 | -22.24 | 2017-2020 |
| Lucinda | 146.39 | -18.52 | 2009-2010;2013-2020 |

