# Peer review of "Long-term multi-source data analysis about the characteristics of aerosol optical properties and types over Australia"

_Atmospheric Chemistry and Physics, 2020_

## Referee Comment (RC1) · Anonymous Referee #1 · 3 Nov 2020

This study uses ground-based and satellite-based remote sensing observations, the reanalysis data, and the HYSPLIT transport model to fully investigate the spatiotemporal distributions of aerosol optical properties and major aerosol types, along with the vertical distribution of major aerosol types over Australia. Valuable and interesting results have been shown, particularly the spatial distributions of aerosol types and temporal variations of aerosol optical depths. These precious findings can help the science community better understand the aerosol characteristics over the Australia regions, which may be potentially used for improvement of both weather and climate models in future. The paper is well organized. I would recommend its acceptance for publication after a minor revision.

General comments: Section 2.2 on pages 6-7. I would recommend the authors make a table to list the data information including the sources, time period, time resolution, and so on. Also, it is necessary and useful for authors to provide the accuracy of the observation data used in this study based on previous literatures.

Specific comments: Line 10, please spell out AERONET for this first time appearance Line 53, "There have been" should change to "There are" Line 54, Since the dry season is not clearly defined, I would suggest the authors directly show the months instead of using "dry season", or they need define the dry season here. Line 63. please just keep one for "In addition" and "also". Line 85-90, some references are needed. Line 92, "it" can be deleted. Line 95, "are exactly" should be "is exactly" Line 102-103, I would suggest mentioning the AOD source using "The uncertianty of the AOD from AERONET is ..." Line 111-112, "Levy et al. (2013) showed that the expected errors of the L2 MODIS AOD product are about  $\pm (0.05+15\%)$ ", it is the MODIS DT AOD, not the data used in this study (i.e., "Deep Blue Aerosol Optical Depth 550 Land Best Estimate"). Please delete this sentence and add the relevant reference. Line 121, "includes" should be "include" Line 124, "AODs" should be used instead of "AOD" Line 160, I wonder what method is used for the trend analysis in this research? Linear analysis? The authors need indicate which method was used. Line 169, "investigated" might be more suitable than "discussed" Line 172, "the size of aerosol particles" is more reasonable Line 188, "zone" should change to "zones" Line 208. "lead to" should be "led to" Line 254. "suggest" should change to "suggested" Line 269, "compared to other sites" should change to "compared to Birdsville" Line 270, delete "the" from "the their" Line 323, "are" should be "were" Line 344, "was" should be "were" Line 430, 'is" should be "was" Page 43, Please enlarge the label in Figure 13(a) Page 44-45, "Seasalt" should be "Sea salt" in Figures 16 and 17

---

## Referee Comment (RC2) · Anonymous Referee #2 · 13 Dec 2020

This paper analyzes the aerosol optical properties, aerosol types, aerosol vertical distribution as well as their spatiotemporal distributions over Australia using observational/reanalysis data from AERONET MODIS, MERRA-2, and CALIOP. This work provides useful information for researchers who are interested in the aerosol properties in Australia. I have the following comments and suggestions which need to be addressed before the paper can be accepted for publication.

1. Abstract: Line 13-16: Please specify what time periods the trends are for. Line 22-26: Please specify which heights these descriptions refer to since you are talking about 0.5-5 km heights in the sentence that follows. Line 26-27: Which region does

this sentence refer to? Line 27-28: The meaning of this sentence is very vague. Line 28-30: This sentence seems to repeat the one in Line 23-26. Please rewritten Line 22-30 to make the meaning clear and avoid overlap.

2. Section 2.2.1: Line 104-106: Does this affect the trend analysis? You may want to do a sensitivity analysis using level 1.5 data for all years and compare with the present results.

3. Section 2.3: Line 146-148: Clarify what data sources you used to determine the aerosol types.

4. Section 3.1.1: Whenever you describe a trend, please specify what time period the trend is for. A trend without a time period is difficult to interpret. Please also describe whether the trends are statistically significant or not. Also, please try to explain why AODs present increasing or decreasing trends at the nine sites. Line 166-167: It is not appropriate to use "trend" for only two years.

5. Section 3.1.2: Please explain why AODs at the nine sites are generally high in spring and summer and low in fall and winter, before explaining why some sites peak in spring and others peak in summer. Line 230: What "marine biogenic emissions" are these? Are they sea-spray aerosols? I doubt whether "biogenic" is an appropriate term here. Line 237: I suggest you avoid using "trend" here. This is confusing since you used "trend" represent interannual trend in the last section.

6. Section 3.2.2: I think a lot of descriptions in the first paragraph are based on Figure 15, but this figure is not cited until the next paragraph. Line 246: Is it really "intercontinental transport"? I think the sentences below do not actually talk about transport from other continents.

7. Section 3.2.3: Please comment on whether the distribution of MERRA-2 based aerosol compositions are consistent with the AERONET based aerosol types.

8. Section 3.3: Whenever possible, please try to make connections between the

CALIPSO based aerosol types and the AERONET based aerosol types used in previous sections. Second paragraph: please clarify that this paragraph describes the vertical profiles averaged across Austria, not one of the three regions defined in the last paragraph.

9. Figure 15: Clarify that the aerosol types are derived from AERONET.

---

## Referee Comment (RC3) · Anonymous Referee #3 · 29 Dec 2020

This study performed a comprehensive analysis of aerosol properties in Australia using multiple observation and reanalysis datasets. It discussed the spatial distribution, trends and seasonality of major aerosol types, combined with an analysis of potential sources using back trajectory simulation. The paper is also well presented and easy to follow. I think this is a good study, offering insights of aerosol variability in Australia, especially the less populated regions of central Australia. I only have a few minor comments and questions:

1. I have a couple of questions regarding the data. In Line 105, I understand that Level 1.5 data from AERONET has much large volume. But it also shows higher uncertainty.

[Figure]

I wonder if any quality control on this data is performed? Actually, keeping all quality screen criteria from AERONET Level 2 algorithm except the AOD threshold will keep most of the data. 2. Section 2.2.2: is there a particular reason that MODIS Terra AOD is not used? 3. Section 2.2.3: the aerosol type analysis is primarily based on MERRA 2 data. I wonder if there is any validation of the MERRA 2 aerosol types, considering that there can be uncertainties in the model simulations? 4. Section 3.1.1: the significance level of all trends should be provided here, especially that most trends are rather small. 5. 3.2.2: I have two questions here. First, I wonder how the back trajectories are clustered? Which method is used? Is it subjective or objective? Second, I think the aerosol source analysis needs to be combined with aerosol type analysis, i.e., what are the potential sources of each aerosol type at each site? For this purpose, I suggest the authors separate the trajectory analysis by aerosol type or by season, according to the results of Figure 15. 6. I am curious about how aerosol properties change during the extremely intense wildfire in late 2019/early 2020. It seems that AOD has greatly increased over Victoria and Australian Capital Territory. Did the authors see other changes in aerosol properties, e.g., AE, absorption, aerosol type, etc? Btw, the location of Australian Capital Territory is not marked on Figure 1. 7. My final comment is that there is lack of a comparison with previous studies. What are the major new findings of the current study as compared with previous studies on aerosol properties in Australia? Is the analysis of aerosol type in this study supported by previous in-situ measurements? Some discussion should be added.

---

## Author Comment (AC1) · 13 Jan 2021

**Reply to Anonymous Reviewer #1:**

**We appreciate the reviewer's comments on the manuscript. All comments are highly valuable and helpful for us to improve our manuscript. We have studied them carefully and have addressed them in the revised manuscript. Below are point-by point responses to the reviewer's comments.**

**Comments from the reviewer:**

**General comments:**

1. Section 2.2 on pages 6-7. I would recommend the authors make a table to list the data information including the sources, time period, time resolution, and so on. Also, it is necessary and useful for authors to provide the accuracy of the observation data used in this study based on previous literatures.

Thank you for the professional suggestion. We have added a table (Table 2 in the manuscript) presenting the detailed information regarding the data information including the sources, dataset name, resolution, and time period.

**Table 2. Summary of datasets used in this study.**

| Instrument/Product | Dataset name | Resolution | Period |
|---|---|---|---|
| AERONET | AOD; AE; SSA | 15 min, Site | 2001.10-2020.5 |
| MODIS | Deep_Blue_Aerosol_Optical_Depth_550_Land_Best_Estimate | Daily, 0.1°×0.1° | 2002.7-2020.5 |
| MERRA-2 | MERRA2_400.tavgM_2d_aer_Nx | Monthly,0.625°×0.5° | 2002.7-2020.5 |
| CALIPSO | CAL_LID_L3_Tropospheric_APro_CloudFree-Standard-V4-20 | Monthly,5°x2° | 2006.6-2020.5 |
| ERA-5 | 10m v-component of wind;10m u-component of wind | Monthly,0.25°x0.25° | 2002.7-2020.5 |
| GDAS | Global Data Assimilation System 1°×1° | Daily, 1°×1° | 2005.1-2020.5 |

Also, we have mentioned the table in the 'study area' section. In addition, we have added descriptions about the accuracy of the data used in the introduction for each dataset.

MODIS:

Line 113-117: "**Sayer et al. (2014) showed that the MODIS Deep Blue (DB) algorithm can retrieve ~1.6 times more AODs than the Dark Target (DT) algorithm with more than 82% of AOD retrievals falling within the expected error envelopes and with small root mean squared error (0.07) over Oceania. As the MODIS Terra and Aqua sensors are near□identical, and the same retrieval algorithms are used to generate AODs, DB algorithm shows very similar performance for the two sensors (Sayer et al.,2013).**"

MERRA-2:

Line 129-132: "**The monthly MERRA-2 and AERONET AOD showed good agreement with correlation coefficients (R) between 0.59 and 0.94 at nine sites over Australia (Fig. S1). Moreover, the root mean squared error (RMSE) were smaller than 0.05, which indicated that the monthly MERRA AOD products showed good performance over Australia.**"

[Figure]

**Figure S1.** The comparisons of monthly MERRA-2 and AERONET AOD at nine AERONET sites over Australia during 2002-2020. Linear regression is shown as a solid red line and all the linear relationships are statistically significant at α = 0.01. The black dashed line is the 1:1 line.

CALIPSO:

Line 139-142: "**Omar et al. (2013) investigated the performance of CALIPSO AOD data and found that when cloud cleared and extinction quality controlled CALIPSO data was compared with AERONET data with AOD less than 1.0, the mean relative difference between the two measurements was 25% of AERONET AOD. In addition, they found that the CALIPSO AOD has good correlation (r=0.65) with AERONET AOD at Lake Argyle in northern Australia**"

**Specific comments:**

2. Line 10, please spell out AERONET for this first time appearance

   Thank you for pointing it out, the correction has been done.

3. Line 53, "There have been" should change to "There are"

   Thank you for your suggestion. We have corrected 'There have been' as ' There are '.

4. Line 54, Since the dry season is not clearly defined, I would suggest the authors directly show the months instead of using "dry season", or they need define the dry season here.

   Thank you for your suggestion. We have defined the period (typically April - November) of the dry season for this first-time appearance.

5. Line 63, please just keep one for "In addition" and "also".

   Thank you for pointing it out, 'also' has been deleted.

6. Line 85-90, some references are needed.

   Thank you for your suggestion. We have added reference in this paragraph. More detail please see the revised manuscript.

7. Line 92, "it" can be deleted.

   Thank you for pointing it out, 'it' has been deleted.

8. Line 95, "are exactly" should be "is exactly"

   It has been changed as 'is exactly'.

9. Line 102-103, I would suggest mentioning the AOD source using "The uncertainty of the AOD from AERONET is ..."

   We have corrected this sentence based on the suggestion.

10. Line 111-112, "Levy et al. (2013) showed that the expected errors of the L2 MODIS AOD product are about ±(0.05+15%)", it is the MODIS DT AOD, not the data used in this study (i.e.,"Deep_Blue_Aerosol_Optical_Depth_550_Land_Best_Estimate"). Please delete this sentence and add the relevant reference.

    We agree with the reviewer. We have deleted the old reference and added new reference to describe the accuracy of MODIS DB AOD in the revised manuscript: "**Sayer et al. (2014) showed that the MODIS Deep Blue (DB) algorithm can retrieve ~1.6 times more AODs than the Dark Target (DT) algorithm with more than 82% of AOD retrievals falling within the expected error envelopes and with small root mean squared error (0.07) over Oceania. As the MODIS Terra and Aqua sensors are near□identical, and the same retrieval algorithms are used to generate AODs, DB algorithm shows very similar performance for the two sensors (Sayer et al.,2013).**" (Line 113-117).

11. Line 121, "includes" should be "include"

    Corrected.

12. Line 124, "AODs" should be used instead of "AOD"

    Corrected.

13. Line 160, I wonder what method is used for the trend analysis in this research? Linear analysis? The authors need indicate which method was used.

    This is a good question. In this study, we used the linear trend analysis to present the trend of AOD and AE over Australia. We have mentioned this method in the revised manuscript in Line 168: "**The annual variations and linear trends of AOD, AE at the nine sites over Australia are shown in Fig. 3.**"

14. Line 169, "investigated" might be more suitable than "discussed"

    We have changed "discussed" as "investigated".

15. Line 172, "the size of aerosol particles" is more reasonable

    Corrected.

16. Line 188, "zone" should change to "zones"

    Corrected.

17. Line 208, "lead to" should be "led to"

    Corrected.

18. Line 254, "suggest" should change to "suggested"

    Corrected.

19. Line 269, "compared to other sites" should change to "compared to Birdsville"

    Corrected.

20. Line 270, delete "the" from "the their"

    Corrected.

21. Line 323, "are" should be "were"

    Corrected.

22. Line 344, "was" should be "were"

23. Corrected.

24. Line 430, 'is" should be "was"

Corrected.

25. Page 43, Please enlarge the label in Figure 13(a)

    The labels have been enlarged.

26. Page 44-45, "Seasalt" should be "Sea salt" in Figures 16 and 17

    Corrected.

---

## Author Comment (AC2) · 13 Jan 2021

**Reply to Anonymous Reviewer #2:**

**We appreciate the reviewer's comments on the manuscript. All comments are highly valuable and helpful for us to improve our manuscript. We have studied them carefully and have addressed them in the revised manuscript. Below are point-by point responses to the reviewer's comments.**

**Comments from the reviewer:**

1.  Abstract: Line 13-16: Please specify what time periods the trends are for. Line 22-26: Please specify which heights these descriptions refer to since you are talking about 0.5-5 km heights in the sentence that follows. Line 26-27: Which region does this sentence refer to? Line 27-28: The meaning of this sentence is very vague. Line 28-30: This sentence seems to repeat the one in Line 23-26. Please rewritten Line 22-30 to make the meaning clear and avoid overlap.

We highly appreciate these detailed comments which have helped us improve the paper quality a lot. We have specified the time period of the trend, and rewritten the Lines 22-30 as suggested, which are in Lines 14-30: "**During the observation period from 2001 to 2020, the annual aerosol optical depth (AOD) at most sites showed increasing trends (0.002-0.029 yr$^{-1}$) except for that at three sites of Canberra, Jabiru, and Lake Argyle, which showed decreasing trends (-0.005 - -0.002 yr$^{-1}$). In contrast, the annual Ångström exponent (AE) showed decreasing tendencies at most sites (-0.045 - -0.005 yr$^{-1}$). The results showed strong seasonal variations in AOD with high values in the austral spring and summer and relatively low values in the austral fall and winter, and weak seasonal variations in AE with the highest mean values in the austral spring at most sites. Monthly averaged AOD increases from August to December or next January, and decreases during the March-July. Spatially, the MODIS AOD showed obvious spatial heterogeneity with high values appeared over the Australian tropical savanna regions, Lake Eyre Basin, and southeastern regions of Australia, while low values appeared over the arid regions in western Australia. The MERRA-2 showed that carbonaceous over northern Australia, dust over central Australia, sulfate over densely populated northwestern and southeastern Australia, and sea salt over Australian coastal regions are the major types of atmospheric aerosols. The nine ground-based AERONET sites over Australia showed that the mixed type of aerosols (biomass burning and dust) are dominant in all seasons. Moreover, the CALIPSO showed that polluted dust is the dominant aerosol type detected at heights 0.5 - 5 km over Australian continent during all seasons. The results suggested that Australian aerosol has similar source characteristics due to the regional transport over Australia, especially for biomass burning and dust aerosols. However, the dust-prone characteristic of aerosol is more prominent over the central Australia, while the biomass burning-prone characteristic of aerosol is more prominent in northern Australia.**".

2.  Section 2.2.1: Line 104-106: Does this affect the trend analysis? You may want to do a sensitivity analysis using level 1.5 data for all years and compare with the present results.

This is a good question. Following the suggestion, we performed a sensitivity analysis between AERONET Daily L1.5 and L2 AOD products, and found that this dataset issue does not affect the trend analysis. As shown in Figure R1, the AOD trends derived from the AERONET Daily L1.5 product are very similar to the trends estimated from combined of Daily L1.5 (2001-2018) and L2.0 (2019-2020) AERONET products (Figure 3 in the manuscript). We have added this information into the manuscript in Line 106-107: "**A sensitivity analysis showed that the use of level 1.5 data**

**should not affect the trend analysis since they show the similar trends for period 2001-2018 when both datasets are available.**"

[Figure]

**Figure R1. Temporal variations of annual mean AOD, AE (Level 1.5) at nine AERONET sites in Australia. Note: "*" means passing the confidence testing at α=0.05.**

3.   Section 2.3: Line 146-148: Clarify what data sources you used to determine the aerosol types.

We have clarified the data sources (i.e., AERONET) which are used to determine the aerosol types by using the two methods from Kaskaoutis et al. (2007) and Giles et al. (2012), which are in Lines 158-160: "**Two methods from Kaskaoutis et al. (2007) and Giles et al. (2012) are adopted to distinguish aerosol types by using the AOD at 500 nm, AE at 440-870 nm, and SSA at 440 nm which are derived from AERONET, as illustrated in Fig. 2.**"

4.   Section 3.1.1: Whenever you describe a trend, please specify what time period the trend is for. A trend without a time period is difficult to interpret. Please also describe whether the trends are statistically significant or not. Also, please try to explain why AODs present increasing or decreasing trends at the nine sites. Line 166-167: It is not appropriate to use "trend" for only two years.

We highly appreciate these suggestions. The time period and significance of trend are now added in the revised manuscript. Moreover, we have tried our best to find and then describe the reasons for the increasing/decreasing trend at the nine sites. We have also removed the sentence regarding the trend for that two years since it is not appropriate. The corresponding changes are in Lines 171-181: "**It showed an increasing tendency in the annual mean AOD at most sites in Australia during the observation period from 2001 to 2020 except for the sites of Canberra, Jabiru, and Lake Argyle, at which the annual mean AOD showed a decreasing trend of -0.004±0.033 yr⁻¹, -0.002±0.035 yr⁻¹, -0.004±0.057 yr⁻¹, respectively. There were no statistically significant trends at most sites during the observation period from 2001 to 2020 except for the sites of Birdsville and Fowlers Gap. The general AOD decrease at sites over northern Australia and the general increase at sites over central Australia are consistent with the findings obtained by Mehta et al. (2016) using MODIS and Multi-angle Imaging SpectroRadiometer (MISR) AOD dataset during the period 2001–2014 and Mitchell et al. (2010) using site observations over the decade 1997–2007. The decreasing annual trends over northern Australia could be associated with the decreases in BC and OC AODs (Yoon et al.,2016).**

**However, increasing AOD trends over central Australia could be mainly attributed to the increase of dust activities (Mitchell et al.,2010).**" and in Lines 185-188: "**It was worth mentioning that significant increases of AOD are observed during the period 2019-2020 at most sites, such as Adelaide Site 7, Birdsville, Fowlers Gap, and Lucinda. The increases in AOD are related to the frequent fire activities in Australia from September 2019 to January 2020.**"

5. Section 3.1.2: Please explain why AODs at the nine sites are generally high in spring and summer and low in fall and winter, before explaining why some sites peak in spring and others peak in summer. Line 230: What "marine biogenic emissions" are these? Are they sea-spray aerosols? I doubt whether "biogenic" is an appropriate term here. Line 237: I suggest you avoid using "trend" here. This is confusing since you used "trend" represent interannual trend in the last section.

We appreciate all of these comments, which are very helpful to us. We have modified our descriptions accordingly based on these comments. In addition, we have removed "marine biogenic emissions" (Line 230) and "trend" (Line 237) since they are not appropriate as the reviewer questioned. The new descriptions are now in Lines 237-259 as follows: "**The main contributors to the high AODs in spring and summer were smoke emissions from biomass burning, dust storms, marine emissions from the sea spray produced in breaking waves, and the natural sulfate particles released from phytoplankton (Rotstayn et al., 2010). The occurrence frequency and intensity of dust storm activities and wildfires decreased in fall and winter, resulting in low AOD values in those two seasons over Australia. Furthermore, the highest seasonal average AOD values were observed in spring at Birdsville (0.09), Jabiru (0.22), Lake Argyle (0.23), and Lucinda (0.13), while they were observed in summer at the other five sites (0.07-0.11). The seasonal variations in AOD observed by AERONET and MODIS were in good agreement at nine sites apart from small differences in magnitude. Mitchell et al. (2013) also found that the AOD values at Wagga and Canberra peaked in summer, while AOD values peaked in spring at sites that are located in the arid zone. This is due to the increasingly forested and bushfire-prone characteristics at the more easterly sites (Mitchell et al., 2013). Moreover, less precipitation and higher wind speeds during spring were observed in northern Australia (north of 18°S), which may lead to the increase in AOD from biomass burning and long-range transport of marine emissions (Fig. 8). During summer, the decrease in AOD was significant over northwestern regions, consistent with the large increase in precipitation and decrease in wind speeds. However, the increase in AOD in eastern and southeastern regions during summer could be associated with the increase of biomass burning and sea salt aerosols that were transported from the Pacific Ocean.**

**The seasonal variation in AE was different from that in AOD. There were no obvious seasonal variations in AE at the nine sites. The maximum seasonal mean AE values (0.92-1.43) were observed in spring at all sites except for Canberra, which was mostly related to the fine particles from biomass burning in spring. Similar results were reported by Mitchell et al. (2013) during the period 1998-2012 in northern Australia. Further, the seasonal mean AE values were greater than 1.0 over all seasons at Canberra, Jabiru, Lake Argyle, Fowlers Gap, and Birdsville, while the mean AE values were less than 1 over all seasons at the Adelaide Site 7 and Luncinda. In addition, at Learmonth and Lake Lefroy, high AE values (0.98-1.07) were observed in spring and fall, and low values (0.56-0.99) were observed in summer and winter.**"

6. Section 3.2.2: I think a lot of descriptions in the first paragraph are based on Figure 15, but

this figure is not cited until the next paragraph. Line 246: Is it really "intercontinental transport"? I think the sentences below do not actually talk about transport from other continents.

The reviewer proposed good questions. We agree with the reviewer and have changed our descriptions by expressing the corresponding figures for our descriptions for the whole section.

It is actually not "intercontinental transport" - we used wrong word since what we want to express is the "long-range transport". We have deleted the wrong description and modified our descriptions at Lines 386-389: "**Figure 15i and Figure S5 showed that air masses originated mainly from the Indian Ocean (>50%), crossing the regions that are affected by wildfires during the spring and summer season, and reaching the Canberra site, which indicated a possible transport of biomass burning and clean marine aerosols from forest regions and ocean, respectively.**".

7. Section 3.2.3: Please comment on whether the distribution of MERRA-2 based aerosol compositions are consistent with the AERONET based aerosol types.

We appreciate this suggestion. We add a description about this at Lines 391-393: "**We should note that the AERONET-based aerosol types show similar spatial distributions as follows by using the MERRA-2 based aerosol types, while they are slightly different in aerosol types classified.**"

8. Section 3.3: Whenever possible, please try to make connections between the CALIPSO based aerosol types and the AERONET based aerosol types used in previous sections. Second paragraph: please clarify that this paragraph describes the vertical profiles averaged across Austria, not one of the three regions defined in the last paragraph.

We appreciate these suggestions and made changes to Section 3.3 accordingly. First, the descriptions about the connections between the CALIPSO based aerosol types and the AERONET based aerosol types are added:

Lines 453-455: "**Further, the aerosol types classified by AERONET over Australia also indicated that the mixed type of aerosols (mostly biomass burning and dust) is the dominant type during all seasons. The result suggested the significant impacts of both biomass burning and desert emissions in Australia.**"

Lines 472-476: "**The results confirmed the existence of dust aerosols in northern Australia, which were mostly generated along with fires and transported from south inland deserts. The altitude with peak occurrence frequency (~5%) for elevated smoke was ~3 km throughout the year. Higher occurrence frequency of elevated smoke was observed at heights from 2 to 4 km in spring, consistent with the result of prevalence of biomass burning aerosol in spring at Lake Argyle, which is located in the biomass burning regime.**"

Second, we clarified that the average vertical profile values are for whole Australia (i.e., the blue shade area in Fig. 18) in Lines 448-449: "**Fig. 19 shows the averaged occurrence frequency profile of each aerosol type in each season from 15-year CALIPSO observations in whole Australia (blue shade region in Fig. 18).**".

9. Figure 15: Clarify that the aerosol types are derived from AERONET

We have revised the caption of Figure 15 (Figure 14 in revised manuscript) to Clarify that the aerosol types are derived from AERONET: "**Relative percentages of different aerosol components in each season at nine sites which are derived from AERONET during the observation period, including dust, biomass burning, mixed, urban/industrial, and uncertain aerosol types.**"

---

## Author Comment (AC3) · 13 Jan 2021

**Reply to Anonymous Reviewer #3:**

We appreciate the reviewer's comments on the manuscript. All comments are highly valuable and helpful for us to improve our manuscript. We have studied them carefully and have addressed them in the revised manuscript. Below are point-by point responses to the reviewer's comments.

**Comments from the reviewer:**

1. I have a couple of questions regarding the data. In Line 105, I understand that Level 1.5 data from AERONET has much large volume. But it also shows higher uncertainty I wonder if any quality control on this data is performed? Actually, keeping all quality screen criteria from AERONET Level 2 algorithm except the AOD threshold will keep most of the data.

   The reviewer proposed a good question. Indeed, Level 1.5 data from AERONET has relatively large uncertainty. Thus, we used level 2.0 data covering most study period (i.e., 2001-2018). However, only five AERONET sites (i.e., Lake Lefory, Learmonth, Jabiru, Lake Argyle, Lucinda), which were located at Australian coastal region, have level 2.0 quality controlled and cloud screened data in 2019. Moreover, only one AERONET site has level 2.0 data in 2020. Hence, there is a lack of level 2.0 data for the central Australia. To obtain a long time series of ground-based observations and to analyze the spatial and temporal characteristics of aerosols over Australia in combination with remote sensing and reanalysis data, we used level 1.5 data for the period 2019-2020 along with the level 2.0 data for the period 2001-2008.

   To make sure whether this combination will affect our trend analysis, we performed a sensitivity analysis between AERONET Daily L1.5 and L2 AOD products, and found that this dataset issue does not affect the trend analysis. As shown in Figure R1, the AOD trends derived from the AERONET Daily L1.5 product are very similar to the trends estimated from combined of Daily L1.5 (2001-2018) and L2.0 (2019-2020) AERONET products (Figure 3 in the manuscript). We have added this information into the manuscript in Line 106-107: "**A sensitivity analysis showed that the use of level 1.5 data should not affect the trend analysis since they show the similar trends for period 2001-2018 when both datasets are available.**".

[Figure]

**Figure R1. Temporal variations of annual mean AOD, AE (Level 1.5) at nine AERONET sites in Australia. Note: "\*" means passing the confidence testing at α=0.05.**

2. Section 2.2.2: is there a particular reason that MODIS Terra AOD is not used?

The reviewer proposed a good question. The reason that we only use one satellite AOD product is that they provide basically the same (very similar) long-term statistical characteristics of AOD (Sayer et al., 2015). Actually, Sayer et al. (2013) showed that AOD errors compared to AERONET appear to be slightly larger for Terra than those from MODIS Aqua by around 3% of the AOD. Considering that we are carrying statistical analysis instead of short-term event analysis, we did not use the MODIS Terra AOD. Of course, when studying short-term AOD characteristics or pollution events, adding MODIS Terra AOD would be much helpful.

Sayer, A. M., Hsu, N. C., Bettenhausen, C., and Jeong, M.-J.: Validation and uncertainty estimates for MODIS Collection 6 "Deep Blue" aerosol data, Journal of Geophysical Research: Atmospheres, 118, 7864-7872, https://doi.org/10.1002/jgrd.50600, 2013.
Sayer, A. M. , Hsu, N. C. , Bettenhausen, C. , Jeong, M. J. , & Meister, G. . (2015). Effect of modis terra radiometric calibration improvements on collection 6 deep blue aerosol products: validation and terra/aqua consistency. Journal of Geophysical Research: Atmospheres.

3. Section 2.2.3: the aerosol type analysis is primarily based on MERRA -2 data. I wonder if there is any validation of the MERRA 2 aerosol types, considering that there can be uncertainties in the model simulations?

This is a good question. It is really challenging for us to evaluate the aerosol types from MERRA-2 without sufficient information. However, considering the potential uncertainties in MERRA-2 data, we have also carried out the aerosol type analysis in Australia by using the combination of AERONET and CALIPSO data in section 3.3, which show very similar results in the spatial distributions of aerosol types and might imply the reliability of the aerosol types from MERRA-2 indirectly.

4. Section 3.1.1: the significance level of all trends should be provided here, especially that most trends are rather small.

We agree with the reviewer. We added significance level information into Figure 3 and revised the corresponding descriptions in Section 3.1.1 in the modified manuscript. More details please see the revised manuscript in Section 3.1.1.

[Figure]

Figure 3. Temporal variations of annual mean AOD, AE at nine AERONET sites in Australia. Note: "*" means passing the confidence testing at α=0.05. "P" means P value.

5. 3.2.2: I have two questions here. First, I wonder how the back trajectories are clustered? Which method is used? Is it subjective or objective? Second, I think the aerosol source analysis needs to be combined with aerosol type analysis, i.e., what are the potential sources of each aerosol type at each site? For this purpose, I suggest the authors separate the trajectory analysis by aerosol type or by season, according to the results of Figure 15.

We appreciate these questions. In this study, we first used the Python + HYSPLIT to generate trajectories during the study period (i.e.,2005-2020). Then we employed the TrajStat module from Meteoinfo version2.4.1 to cluster the back trajectories (http://meteothink.org/docs/trajstat/cluster_cal.html). There are two clustering options with Euclidean distance or angle distance. In this study, we used the Euclidean distance method for cluster Calculation. Moreover, the Total spatial variation (TSV) was calculated to determine the class number of back trajectories. Thus, the cluster method is objective. The TSV percent change vs cluster numbers figure can indicate the suitable cluster number before dramatic increasing of TSV percent. More detail information please see the introduction of the TrajStat module (http://meteothink.org/docs/trajstat/cluster_cal.html). We have briefly indicated the cluster method in our method part in Section 2.3 at Lines 166-167: "Note that we have employed the TrajStat module from Meteoinfo version2.4.1 to cluster the back trajectories by using the Euclidean distance method (http://meteothink.org/docs/trajstat/index.html)."

Second, we have separated the trajectories by season based on your suggestion, with the Figures shown in the supplementary (Figs. S2-S5). Following your suggestion, we have added the relevant conclusions in our revised manuscript at several parts. For example,

Line 344-348: "**Furthermore, the seasonal trajectories evidenced the dust aerosols transport from the southeastern deserts (Fig. S2). The results were similar to the findings of McGowan and Clark (2008), who also demonstrated that dust transport from Lake Eyre could travel through the northwest dust transport pathway to affect the northern Australia, Indonesia and the southern Philippines by using HYSPLIT model during 1980-2000.**".

Line 369-374: "**Seasonal trajectories showed that 50% of the airflow at Birdsville was from the eastern and southeastern Australia during four seasons, which evidenced the biomass burning aerosols transport, especially during spring and summer (Fig. S4). This was supported by the findings of Qin et al. (2009), who demonstrated that who demonstrated that smoke generated from fires in Canberra could be transported over 1500 km across New South Wales to the central Australia.**".

Line 386-389:"**Figure 15i and Figure S5 showed that air masses originated mainly from the Indian Ocean (>50%), crossing the regions that are affected by wildfires during the spring and summer season, and reaching the Canberra site, which indicated a possible transport of biomass burning and clean marine aerosols from forest regions and ocean, respectively.**".

[Figure]

Figure S2. Cluster analysis of simulated back trajectories from HYSPLIT during the period January 2005-May 2020 for air masses ending at Jabiru and Lake Argyle at 500 m above ground level in Australia.

[Figure]

Figure S3. Cluster analysis of simulated back trajectories from HYSPLIT during the period January 2005-May 2020 for air masses ending at Learmonth and Lake Lefory at 500 m above ground level in Australia.

[Figure]

Figure S4. Cluster analysis of simulated back trajectories from HYSPLIT during the period January 2005-May 2020 for air masses ending at Birdsville, Fowlers Gap, and Adelaide Site 7 at 500 m above ground level in Australia.

[Figure]

Figure S5. Cluster analysis of simulated back trajectories from HYSPLIT during the period January 2005-May 2020 for air masses ending at Lucinda and Canberra at 500 m above ground level in Australia.

6. I am curious about how aerosol properties change during the extremely intense wildfire in late 2019/early 2020. It seems that AOD has greatly increased over Victoria and Australian Capital Territory. Did the authors see other changes in aerosol properties, e.g., AE, absorption, aerosol type, etc? Btw, the location of Australian Capital Territory is not marked on Figure 1.

The reviewer proposed a very good point. Yes, we did see the changes in aerosol properties during this extreme intense wildfire event. The extremely intense wildfire during the 2019/2020 fire season had significant impact on aerosol properties, such as the extreme increase in AOD for most southeastern Australia, the dominance of fine particle aerosols, and the significant increase in carbonaceous and dust aerosols in southeastern and central Australia, respectively. We are making a comprehensive special investigation about the optical and physical properties of aerosols during the Australia wildfires in 2019/2020 following this study, and would be more than happy to provide

you those results in future.

In Figure 1, we now add the label of Australian Capital Territory.

7. My final comment is that there is lack of a comparison with previous studies. What are the major new findings of the current study as compared with previous studies on aerosol properties in Australia? Is the analysis of aerosol type in this study supported by previous in-situ measurements? Some discussion should be added.

We appreciate this valuable comment. In most of previous studies, fire season has often been the focus study period due to the significant impacts of biomass burning aerosol. Meanwhile, most of previous studies have focused on aerosol properties at a specific site/region or short-term variations of aerosols due to the difficulty of obtaining ground-based aerosol data. This study analyzed the long-term spatial and temporal properties of aerosols over Australia from ground-based measurements (i.e. AERONET), remote sensing observations (MODIS, CALIPSO), and reanalysis data. The major new findings are about the long-term trends of AOD and AE, the spatio-temporal variations of AOD, and the vertical distributions of aerosol optical properties.

Following this suggestion, we have added the discussions by comparing with previous findings in our revised manuscript at several parts. For example,

Line 342-343: "**Karlson et al. (2014) analyzed the element and particle size of dust deposited in northwestern Australia from 2008 to 2009 and found that samples of Halls Creek (located in the southwest of lake argyle) were derived from the central Australia (e.g., the Lake Eyre Basin).**".

Line 371-373:"**This was supported by the findings of Qin et al. (2009), who demonstrated that smoke generated from fires in Canberra could be transported over 1500 km across New South Wales to the central Australia.**"

Line 382-384: "**Many previous studies also showed that urban/industrial aerosols and biomass burning aerosols were the main components of aerosols at Canberra (Qin et al., 2009; Mitchell et al., 2006; Provençal et al., 2017). Moreover, clean marine and dust aerosols were abundant during fall and winter, which was associated with its location in coastal areas and southeast dust transport corridors (McGowan and Clark., 2008).**".

---

## Author Response (AR2)

**Response to referees on "Long-term multi-source data analysis about the characteristics of aerosol optical properties and types over Australia" by Xingchuan Yang et al.**

Xingchuan Yang[1], Chuanfeng Zhao[1]*, Yikun Yang[1], Hao Fan[1]

1. State Key Laboratory of Earth Surface Processes and Resource Ecology, and College of Global

Change and Earth System Science, Beijing Normal University, Beijing, China

**We would like to thank the editor for carefully reading the manuscript and providing valuable suggestions to improve our manuscript quality. We have addressed them in the revised manuscript, which includes additional investigations. For clarity, the comments below are in black and our responses are in blue.**

**Comments from the editor:**

1. My comment for the paper before publication is that the paper is a bit too long with 21 figures. Some figures (e.g., Fig 8 for precipitation) can be put in the supplement, and only leave the most essential figures and results in the main text.

   We appreciate this valuable comment. We have reduced the number of figures in the previous manuscript from 21 to 17 in the revised manuscript. The main changes are as follows.

   (1) We put Figure 8 of Section 3.1.2 in the supplement as Fig. S2.

   (2) We put Figure 18 of Section 3.3 in the supplement as Fig. S7.

   (3) We merged Figures 19, 20 and 21 from the original manuscript into one figure, and put in the main text as Figure 17 in the revised manuscript.

   We also made corresponding changes in the main text according to the serial number of the figures.

[Figure]

Figure 17. Occurrence frequency profile of each aerosol type in four seasons from 15-year CALIPSO L3 aerosol profile data product in Australian continent (a-d), the biomass burning regime (e-h), and the desert regime (i-l).